



# Using ERA-Interim Reanalysis output for creating datasets of energy-relevant climate variables

Philip D. Jones[1,4], Colin Harpham[1], Alberto Troccoli[2], Benoit Gschwind[3], Thierry Ranchin[3], Lucien Wald[3], Clare M. Goodess[1] and Stephen Dorling[2]

[1]Climatic Research Unit (CRU), School of Environmental Sciences, University of East Anglia, Norwich, NR4 7TJ, UK
[2]School of Environmental Sciences, University of East Anglia, Norwich, NR4 7TJ, UK
[3]MINES ParisTech, PSL Research University, O.I.E. - Centre Observation, Impacts, Energy, 06904 Sophia Antipolis, France
[4]Center of Excellence for Climate Change Research, Department of Meteorology, King Abdulaziz University, Jeddah, Saudi Arabia

*Correspondence to*: Philip D Jones (p.jones@uea.ac.uk)

**Abstract.** The construction of a bias-adjusted dataset of climate variables at the near surface using ERA-Interim Reanalysis is presented. A number of different bias-adjustment approaches have been proposed. Here we modify the parameters of different distributions (depending on the variable), adjusting those calculated from ERA-Interim to those based on gridded station or direct station observations. The variables are air temperature, dewpoint temperature, precipitation (daily only), solar radiation,

wind speed and relative humidity, available at either 3 or 6 h timescales over the period 1979-2014. This dataset is available to anyone through the Climate Data Store (CDS) of the Copernicus Climate Change Data Store (C3S), and can be accessed at present from (ftp://ecem.climate.copernicus.eu ). The benefit of performing bias-adjustment is demonstrated by comparing initial and bias-adjusted ERA-Interim data against observations.

## 1 Introduction

Climate/weather information has been widely used in a number of climate-related impact sectors (e.g. agriculture, water and energy) for decades. Increasingly, users are moving beyond the use of station observations to the use of gridded products, especially meteorological reanalysis datasets. These are reconstructions of past climates produced through the blending of observations with physical/numerical models which have been developed explicitly for climate monitoring and research (Compo et al., 2011; Dee et al., 2011; Hersbach et al., 2015). Reanalyses have the specific advantage of being complete through

the process of physical/dynamic representation of the climate system which provides internally consistent fields across most surface atmospheric variables as well as in the atmospheric column up to the stratosphere (Compo et al., 2011). The present paper deals with the use of Reanalysis for the production of datasets of climate variables relevant to energy. The work took place within the European Climatic Energy Mixes (ECEM) project in the framework of the Copernicus Climate Change Service (C3S) Sectoral Information Service (SIS). This project is primarily focused on users in the energy sector who are interested in

sub-daily (e.g. 6 h) and daily variability for the following variables at the near surface: air temperature, dewpoint temperature,



precipitation, solar radiation, wind speed and relative humidity. Despite this choice of variables being of primary relevance to the energy sector, it is likely that the results will also be of use to other sectors (particularly water).

Because reanalyses are computed on a model grid, inevitably there will be differences when compared to station observations.
Differences are not solely related to scales: Reanalyses are dependent on the underlying climate model and the amount of observational data entering the assimilation system used to produce them. Many users of Reanalysis products attempt to adjust them to observational distributions through a process that is referred to using different terminology: bias adjustment and calibration being the most commonly used terms (Maraun et al., 2010). Here, we use the term bias-adjustment.

The principal reason for performing a bias-adjustment is that reanalyses are potentially biased compared to direct station observations (even when the station observations are gridded to a comparable spatial resolution), more so for some variables than others (e.g. precipitation compared to temperature) and the bias may also vary in value and space, i.e. the bias may be larger for more extreme values and it might be larger for regions of sparse station coverage. The importance of the bias depends to a large extent on how the data will be used. For some variables, the monthly average/totals will be important, but many
other users require that extremes of the distribution be well simulated. With time the complexity of approaches to bias adjustment has developed from getting the monthly averages correct to the present attempts to adjust the whole distribution and to even account for the multivariate relationships between some variables (see e.g. Vrac and Friederichs, 2015). These advances reflect not only the greater expectations with each generation of Reanalysis, but also the greater number of users in a greater number of sectors.

Bias adjustment in the WATCH project (Weedon et al., 2011, 2014) was undertaken at the monthly average scale, with no multivariate dependence and for a number of hydrological variables necessary to calculate evapotranspiration, soil moisture and runoff (so including air temperature, precipitation, long-wave and short-wave/solar radiation, wind speed, specific humidity and surface pressure) and for the period of analysis 1958-2001 (1979-2013) based on the ERA-40 (ERA-Interim)
Reanalysis. The spatial coverage is all land areas north of latitude 60°S. ECEM is less spatially extensive than the WATCH project. It covers the European Domain (27°N-72°N, 22°W-45°E). The current period of study is 1979-2014 based on the ERA-Interim Reanalysis with sub-daily and daily timescales.

The aim of this paper is to present the construction of a bias-adjusted dataset of the climate variables listed above, by using
ERA-Interim Reanalysis. This dataset is available to anyone through the CDS of C3S (currently ftp://ecem.climate.copernicus.eu ) . The benefit of performing bias-adjustment is demonstrated by comparing initial and bias-adjusted data against station observations and gridded observation products. The ERA-Interim Reanalysis and the gridded and station observation-based datasets used for bias adjustment are described in Section 2. Section 3 provides more information on the motivations for bias adjustment, together with some discussion on a number of proposed methods. As we are proposing



bias adjustment at the daily and sub-daily timescales, we will consider whether we need multivariate schemes (as bias adjusting one variable may impact others), particularly whether this is essential in our Energy Sector context. Our selected techniques are discussed in Section 4. In Section 5, we also briefly discuss issues related to whether our bias adjustment is applicable to other sectors, or whether there is any technique that could be applicable across all possible user sectors. Different sectors have

different user demands relating to variables required, timescales and the length of historical Reanalysis data needed. Section 6 provides some conclusions.

**2 Data**

This section provides details of ERA-Interim, and the various gridded and station observation datasets used to assess the quality of this Reanalysis. With gridded datasets, the spatial resolutions may vary, so this section includes how a dataset may be

regridded and also how a station dataset can be interpolated to a grid, if necessary.

**2.1 ERA-Interim**

The development of ERA-Interim is described by Dee et al. (2011). Surface air temperature, precipitation, wind speed at 10 m, surface solar irradiance and relative humidity were extracted from ERA-Interim on the native grid (~0.7° by 0.7° latitude/longitude grid). The period is 1979-2014 and the temporal resolution is either 3 h (forecast) or 6 h (analysis), depending

on the variable (see Dee et al., 2011 for details). These five are Essential Climate Variables (ECVs) defined by the Global Climate Observing System (Bojinski et al., 2014). After extraction, the variables have been spatially interpolated to a more user-friendly 0.5° by 0.5° grid for the ECEM domain, by regridding ERA-Interim using a bi-linear interpolation technique. There are two principal reasons for this regridding: *i)* some of the observation datasets for the assessment of ERA-Interim are available on this regular latitude/longitude grid and *ii)* potential users of the datasets developed here requested regular

latitude/longitude grids with cells size of 0.5° for practical reasons (in particular for aggregation to the country scale). It is also preferable to regrid a dataset without missing values, as opposed to an observation-based gridded product as these can contain missing values when some station data were not available.

**2.2 Gridded observation datasets**

Among the available gridded products for air temperature and precipitation, we used:

• E-OBS for both variables (http://www.ecad.eu/download/ensembles/ensembles.php, Haylock et al., 2008)
     • CRU for both variables (CRU TS 3.23, https://crudata.uea.ac.uk/cru/data/hrg/, Harris et al., 2014) and
     • GPCC (Global Precipitation Climatology Centre) for precipitation
       (https://www.dwd.de/EN/ourservices/gpcc/gpcc.html, Becker et al., 2013).

E-OBS, CRU and GPCC data were downloaded for the ECEM grid. All three datasets only cover land regions, so any bias

adjustment using these data sets will not include marine areas. E-OBS covers the period from 1951-2015, so fully



encompassing the 1979-2014 period of ERA-Interim. CRU TS and GPCC cover the period from 1901 to the present, but are both monthly averages/totals, so can only provide an assessment at this timescale. They are of potential use in regions of the domain where daily timescale data are sparse and/or often missing in E-OBS (e.g. North Africa and the Middle East). The principal emphases of ECEM are the countries of Western Europe, so North Africa and parts of the Middle East are often
missing on some of the subsequent maps.

### 2.3 HadISD

No gridded observed product is available for wind speed and dewpoint temperature. Dewpoint temperature is necessary as it can be combined with air temperature to calculate relative humidity, which is needed for energy calculations, such as demand. Station data for wind speed at 10 m height and dewpoint temperature were extracted from HadISD
(http://www.metoffice.gov.uk/hadobs/hadisd/) for approximately 1500 stations across Europe. Station data were extracted every 6 h at the SYNOP hours 00, 06, 12 and 18 for the period 1979 to 2014. HadISD has been assessed for long-term homogeneity by Dunn et al. (2014). Variations in station coverage within HadISD are considerably greater than the coverage achieved for air temperature from E-OBS and precipitation from E-OBS and GPCC. This indicates that it would be unwise to attempt spatial interpolation to a 0.5° by 0.5° grid. Instead each station series will be compared with that from the nearest ERA-
Interim grid box series.

### 2.4 Surface solar irradiance from the World Radiation Data Center

National Meteorological Services (NMS) usually measure surface solar irradiance at a few sites. Data are sent to the World Radiation Data Center (WRDC), a laboratory of the Voeikov Main Geophysical Observatory in Saint-Petersburg, Russia, under the control of the World Meteorological Organization (WMO). There, the data are archived and published
(wrdc.mgo.rssi.ru). Most of the data are daily irradiation; hourly irradiation is available at very few sites. All data are scrutinized at WRDC and quality-flagged before entering archives. 57 stations with high-quality daily irradiation data were kept, for which mean daily irradiance was computed.

### 2.5 HelioClim-3v5 (HC3v5)

Boilley and Wald (2015) have shown the need to correct ERA-Interim estimates of solar irradiance. As only few stations are
available for solar irradiance, it was decided to exploit satellite-derived datasets to correct ERA-Interim. HelioClim-3v5 (HC3v5) is such a dataset and results from the long-standing effort by MINES ParisTech for providing accurate assessment of surface solar irradiance (Blanc et al., 2011). HC3v5 originates from the daily processing of images acquired by the series of satellites Meteosat-MSG and as such does not cover the extreme northern part of the ECEM domain. The first estimates began on 1st February 2004 and these have been compared satisfactorily with measurements taken at ground stations (Eissa et al.,
2015; Thomas et al., 2016a, b). HC3v5 data were downloaded from the SoDa Service web site (www.soda-pro.com) from



which one may select the timescale, here daily mean of irradiance $I$. The HC3v5 product comprises the irradiance at the top of atmosphere $E_0$ from which one may compute the clearness index $KT$:

$$KT = I/E_0 \qquad\qquad\qquad (1)$$

$KT$ is a good indicator of the optical state of the atmosphere with a dependency on the position of the sun much less pronounced

than in $I$. $KT$ close to 0.7-0.8 signifies a clear sky, while $KT$ close to 0.2 signifies an overcast sky.

### 3. Bias-adjustment approaches

Bias adjustment and bias correction are widely-used terms for the assessment of climate model output (from both Global and Regional Climate Models, GCMs and RCMs, see e.g. Maraun et al., 2010 and Maraun, 2012) generally through comparison

with station observational data. In this context, the biases are often much larger than differences with recent Reanalysis products. There are a number of studies where GCMs and RCMs are bias adjusted against Reanalyses, so the assumption is made there that Reanalyses are correct. This happens more in regions where observational datasets are sparse and/or hard to access. Bias adjustment of Reanalyses has been undertaken for a number of years, though. An extensive exercise was carried out by the WATCH project (http://www.eu-watch.org/, see Weedon et al. 2011, 2014). These used the CRU TS dataset as the

basis for adjusting ERA-40 and ERA-Interim, and the adjustments are based on average monthly differences treating each variable independently from each other.

Numerous and more complex (than Weedon et al., 2014) methods for bias adjusting climate variables derived from climate models have been proposed. A number of review papers have been published (e.g. Maraun *et al*., 2010, Maruan, 2013 and

Vrac and Friedrichs, 2015). Among the various possibilities are  the cumulative distribution function (CDF) transform method of Vrac et al. (2012), the distribution based scaling (DBS) method of Yang et al, 2010, empirical quantile mapping (Themeßl et al., 2011, 2012; Wilcke et al., 2013) and using the R package 'qmap' used by MetNorway (Gudmundsson et al., 2012). Unlike the bias adjustment within the WATCH project, the latest examples from the literature attempt to address the issues of spatial dependence of the bias (any bias in ERA-Interim for a variable is expected to be relatively smooth) and temporal

dependence (biases may be greater for certain types of weather, which has led to the approaches improving the fit between the distributions). Also some of the latest techniques attempt to adjust climate variables in a multivariate way. An issue not often addressed is whether the bias varies with time, but the quality of ERA-Interim and also our gridded observational series are both likely to slightly vary as well, so this is very difficult to assess. If trends in biases are evident should adjustment only occur where the difference trends are statistically significant?

Research in the literature has tended to emphasize precipitation (where bias adjustment can also be classed as a form of downscaling). In ECEM, precipitation is less important, with instead a greater emphasis on wind speed and solar irradiance as well as temperature. Most recent work (e.g. Vrac and Friedrichs, 2015) has also addressed multivariate adjustment (changes





in one variable will affect others), but most studies only address pairs of variables rather than three or more. Also the pair usually considered is temperature and precipitation, whereas for other variables relevant to energy work is still at an embryonic stage. As stated earlier, how good bias adjustment has to be depends on how the adjusted data will be used. Within ECEM, we need the techniques to be fit for purpose and that purpose is the ECEM project. Within the project, we will use up to five

variables, so is multivariate bias-adjustment at this scale possible? Even though our users in the Energy Sector are a diverse group, they are mainly interested in only one or two variables, and our initial determination of their needs indicated that univariate bias adjustment will be sufficient.

## 4 Bias-adjustment and results

In the present work, we began by comparing ERA-Interim against the gridded observational products at the monthly timescale,

essentially the same approach as Weedon et al. (2014). The bias was computed as the mean of the differences (model minus observations). For both temperature and precipitation (not shown), differences are generally greater (but variable in sign) over mountainous regions and some coastal areas (the Norwegian coast for temperature and most west-facing coasts for precipitation). Energy sectors users are much more interested in the extremes of the distribution, so our approach moved to adjusting the whole ERA-Interim distribution, using a different statistical distribution for each variable. We begin with wind

speed, then move to air and dewpoint temperature, then precipitation and finally a new approach entirely for solar radiation.

## 4.1 Wind speed at 10 m

In this section results from our univariate bias adjustment are presented starting with wind speed at 10 m. For use in the energy sector, wind speeds at hub heights (80-120 m) are potentially more useful, but assessing ERA-Interim wind speeds from these heights is only possible at a limited number of masts. Assessment over the whole domain is only possible using surface station

measurements which measure wind speeds at 10 m. The two-parameter Weibull distribution is the most-used probability distribution for representing wind speeds and is of strong relevance in the energy sector. The Weibull distribution, with scale parameter $\alpha{>}0$ and shape parameter $\beta{>}0$, has a cumulative distribution function for $x{>}0$ given by:

$$\Pr(X \leq x) = F(x; \alpha, \beta) = 1 - \exp\left[-\left(\frac{x}{\alpha}\right)^{\beta}\right] \tag{2}$$

The scale parameter $\alpha$ relates to the mean wind speed and $\beta$ characterizes the skewness of the distribution; typical values of $\beta$

range between 1 (highly variable wind speed) and 3 (fairly constant wind speed). The 2-parameter Weibull distribution was fitted to 6-h wind speed data from ERA-Interim on a monthly basis, i.e. a separate fit was made for each month, for each grid box using all the 6-h data for 1981-2010, irrespective of the wind direction. The same approach was applied for the wind data from 803 stations in the HadISD dataset that have at least 66.6% data completeness for this 30-year period. The shape and scale parameters ($\alpha$, $\beta$) for the 803 stations were compared with the same parameters from the nearest ERA-Interim grid box.



Figure 1 shows differences (ERA-Interim minus observations) between the scale and shape parameters for January across the European domain. The maps indicate generally good agreement for January, i.e. the values for the two parameters are within ±1 of each other. Exceptions may be found in some mountainous regions and around west-facing coasts but this is very dependent on the month (larger differences when wind speeds are stronger). The similarity of the two distributions in terms of

their scale and shape parameters indicates that bias adjustment could be achieved by replacing the ERA-Interim scale and shape parameters with those inferred from the HadISD stations.

Equation 3 of Tye et al. (2014) provides a means to adjust the original variable $X$ into a variable $X^*$ having scale and shape parameters $\alpha^*$ and $\beta^*$ by the following power-law transfer function:

$$X^* = \alpha^* \left(\frac{X}{\alpha}\right)^{\beta/\beta^*} \tag{3}$$

Where stations are available, $\alpha^*$ and $\beta^*$ are those of the stations. The scale and shape parameters computed at stations were interpolated to each ERA-Interim grid box with the bi-linear INTERP function within the R Akima software package. A bias-adjusted dataset of wind speeds for ERA-Interim is obtained by applying Eq 3. Figures 2 and 3 exhibit the Weibull distribution fits of the HadISD observations, original ERA-Interim and bias-adjusted ERA-Interim for the twelve calendar months for the stations Kirkwall, Scotland, and Maribor, Slovenia. These two locations were chosen as one is maritime and the other more

continental. The other 801 distributional fits are shown on the website with the unadjusted and adjusted ERA-Interim grids (ftp://ecem.climate.copernicus.eu). It is clear from these two examples that the Weibull distributional fit for the stations has moved the adjusted ERA-Interim data series towards the observational distribution, more so for Kirkwall which shows a much greater improvement than for Maribor, where the distribution moves are a clear improvement in winter months but less so for spring and early summer months. Bias adjustment is less successful than the examples shown for a few stations located in

coastal areas and a few sites in mountainous regions. Some observed distributions are a little erratic due to some years in the observed data having wind speeds rounded to integer values. Similarly to Figure 1, but for bias-adjusted ERA-interim minus observations, Figure 4 shows differences between the scale and shape parameters for January. Most stations across Europe exhibit similar shape and scale parameters between the stations and ERA-Interim. However, a few stations in coastal areas and at high elevation mountain locations still show differences in parameters. The fit is not perfectly attained as estimation of the

shape and scale parameters for the ERA-Interim grid boxes from HadISD is influenced by the station distribution. In addition, the number of stations in some parts of Europe is less dense, so involving greater extrapolation from stations more distant from the grid boxes.

### 4.2 Surface air temperature, dewpoint temperature and relative humidity

Like wind speed, both surface air temperature and dewpoint temperature are produced from ERA-Interim every 6 h. Unlike

10 m-wind, both these variables have a strong diurnal cycle, which is generally slightly stronger in the summer. A normal distribution was fitted using daily averages of temperature, taking the average of the four 6-h data for each day. E-OBS is the





dataset on which ERA-Interim is to be adjusted for air temperature, while for dewpoint we use HadISD in a similar way to wind speeds. Means and standard deviations of daily average of temperature are calculated for each month of the year for each 0.5° grid cell of ERA-Interim and for the nearest E-OBS grid box, and for HadISD stations. The means and standard deviations of HadISD stations are then interpolated as for wind speed. Data are then normalized as in Equations 4 and 5.

$$T'_{ERA} = \frac{T_{ERA} - \bar{T}_{ERA}}{\sigma_{ERA}} \tag{4}$$

$$T^* = T'_{ERA} \, \sigma_{obs} + \bar{T}_{obs} \tag{5}$$

where $T'$ is the normalised ERA-Interim temperature anomaly, $T^*$ is the bias-adjusted ERA-Interim temperature, $\bar{T}$ is the mean temperature and $\sigma$ is the standard deviation. Bias adjustment works by transforming the normalized ERA-Interim grid-box time series back to air temperatures using the means and standard deviations from E-OBS and interpolations from station data in HadISD for dewpoints. Once daily averages are adjusted, the difference between the original ERA daily mean and the adjusted daily mean is added to each of the four 6-h temperatures within each day. Therefore, no alteration is made to each diurnal cycle of air or dewpoint temperature. This yields the final set of bias-adjusted 6-h surface air and dewpoint temperatures. It might seem more logical to use the same temporal resolution dataset for both temperatures (e.g. by using air temperatures from HadISD), but this would involve not taking advantage of E-OBS, which uses far more input station data than HadISD.

Figure 5 shows the differences in the mean and standard deviation for air temperature for April as an example. There is good agreement between estimates for ERA-Interim and those calculated from E-OBS. As these are both gridded datasets, the maps shown are fully coloured for each 0.5° grid box. Figures 6 and 7 exhibit the normal distribution fits of the E-OBS, original ERA-Interim and bias-adjusted ERA-Interim for the twelve calendar months for the nearest land grid boxes that approximate the locations of Kirkwall and Maribor used for wind speed. For Maribor this is the 0.5° grid box where the city is located. Kirkwall is on the Orkney Islands, so the nearest grid box within E-OBS is located further south in Northern Scotland. The distributional fits for the Maribor grid box were good for ERA-Interim and bias adjustment brings minor improvement. For the Kirkwall grid box, the adjustments improve the fits in all months, but are less good for the cold tail of air temperature in winter. The full set of results for the 4621 grid-box comparisons can be viewed on the website. Similarly to Figure 5, Figure 8 shows the differences between the means and standard deviations, but this time after adjustment.

Figure 9 shows the differences in the mean and standard deviation for dewpoint temperature for July. There is good agreement between estimates from ERA-Interim and those calculated from HadISD. This plot shows the station locations in a similar fashion to that for wind in Figure 1. Figures 10 and 11 show the normal distribution fits of the HadISD, original ERA-Interim and bias-adjusted ERA-Interim for the twelve calendar months for the locations of Kirkwall and Maribor. Both examples of



distributional plots adjust ERA-Interim slightly, but the original fits were quite good to start with. Similarly to Figure 9, Figure 12 shows the differences between the means and standard deviations but after adjustment.

With the adjustments for dewpoint temperature, it is a simple task to then calculate relative humidity (RH) using the adjusted
air temperature. A small percentage of values (5.4%) exhibits adjusted dewpoint temperature greater than the adjusted air temperature; in these cases, RH was set to 100%. Independently bias-adjusting dewpoint and air temperature is the likely cause of this issue, as this is not the case with the original ERA-Interim data. The majority of the cases where dewpoint exceeded air temperature occur in northern Europe in winter months and in mountainous regions. Any type of bias adjustment procedure will additionally be influenced by the quality of the station observations, and especially the times of observations as E-OBS
air temperatures are daily maximum and minimum temperatures and HadISD are 6-h dewpoint temperatures, and also by the large differences in potential height between some observing locations and the average height field used by ERA-Interim.

**4.3 Daily precipitation totals**

The same process was then used for daily precipitation totals, but using a gamma distribution, which has been found to perform well in many studies (e.g. Wilks, 1995).

$$\Pr(X \leq x) = F(x; \alpha, \beta) = \left(\frac{x}{\beta}\right)^{\alpha-1} \frac{\exp\left(-\frac{x}{\beta}\right)}{\beta \, \Gamma(\alpha)}$$
      (6)

Gamma distributions have two parameters, scale (α) and shape (β), and were fit to the daily precipitation totals for each month for ERA-Interim and for E-OBS. In Equation 6, Γ is the gamma function. This approach to bias adjustment has been used by Piani et al. (2010). We experimented with using or ignoring all precipitation values below a fixed low daily precipitation
threshold over the whole domain. Thresholds of 0.4, 0.6, 0.8 and 1.0 mm were experimented with and best fits were achieved with 1.0 mm. This implies that the gamma distributional fits are based only on days with precipitation values greater than the threshold, with a different fit for each month. This threshold ignores small precipitation totals, more so for ERA-Interim than for E-OBS, but as both datasets are in essence areal averages, more than would be the case for a station rain gauge series. In the adjusted ERA-Interim all precipitation amounts below the threshold are set to zero, further improving the agreement
between E-OBS and ERA-Interim in the number of dry days per month (i.e. days with rainfall less than the 1.0 mm threshold). Adjustment is performed in a similar way to the temperatures, by back transforming the transformed ERA-Interim precipitation total with the scale and shape parameters from the E-OBS dataset.

Figure 13 shows the differences in the scale and shape parameters of the gamma distribution for October, by way of example.
There is good agreement between estimates for ERA-Interim and those calculated from E-OBS. As these are both gridded datasets, the maps shown are fully coloured for each 0.5° grid box. Of the 4520 possibilities, Figures 14 and 15 exhibit the gamma distribution fits of the E-OBS, original ERA-Interim and bias-adjusted ERA-Interim for the twelve calendar months





for the nearest land grid boxes that approximate the locations of Kirkwall and Maribor. The fits for the northern Scotland grid box are considerably better than for Maribor. The full set of results for the 4520 grid-box comparisons can be viewed at the website. Although the gamma distribution is widely used for rainfall data, it is not ideal in all climates and across all seasons in Europe. Problems arise when there are too few rainfall days within dry seasons (the Southern Mediterranean and the Middle

East during summer). Similarly to Figure 13, Figure 16 shows the differences between the scale and shape parameters after adjustment.

**4.4 Surface solar irradiance**

For the sake of simplicity, the adjustment was performed on the daily mean of irradiance. Three methods have been investigated: ratio, affine and quantile mapping. The method 'ratio' consists of computing the means of HC3v5 $\overline{I_{HC3v5}}$ and

ERA-Interim $\overline{I_{ERA}}$ for the calibration period: 2005-2014, then computing the ratio of these means ($\overline{I_{HC3v5}}/\overline{I_{ERA}}$) and eventually multiplying the ERA-Interim estimates by this ratio for the entire period. The method 'affine' consists in adjusting an affine function between HC3v5 and ERA for the calibration period and then applying this function to the ERA-Interim estimates. The method 'quantile mapping' consists of adjusting the cumulative distribution function of ERA-Interim onto that of HC3v5 for the calibration period, thus yielding an abacus that is used to convert the ERA-Interim estimates into adjusted irradiances.

Each method may be applied to the clearness indices $KT$ as well. The possible improvement in bias delivered by each method was assessed by comparing the original ERA-Interim estimates and the bias-adjusted ERA-Interim with measurements from the 57 WRDC stations. The method 'quantile mapping' applied to $KT$ was preferred (and is used here) as it usually brings improvement with no degradation of the bias, while the other methods often degrade the bias in a noticeable way.

Figure 17 exhibits the bias for ERA-Interim vs ground observations of daily mean of solar irradiance for the 57 stations. Downward triangles mean a negative bias of more than -5 W m$^{-2}$, upward triangles mean a positive bias greater than 5 W m$^{-2}$ and circles mean an absolute value of the bias less than 5 W m$^{-2}$. The size of the triangles increases with increasing absolute value of the bias. Bias is often positive: i.e., ERA-Interim tends to overestimate the surface solar irradiance. Only 12 stations out of 57 exhibit an absolute bias of less than 5 W m$^{-2}$.

HC3v5 does not cover latitudes north of 60°N. Two stations: Lerwick (Scotland) and Borlange (Sweden) are located along this latitude (Fig. 17). No adjustment is performed to the grid boxes which are outside the coverage of HC3v5, except for the grid cells along the border where the new irradiance values are set to the mean of the original and adjusted irradiances to avoid spatial discontinuities. Figure 18 exhibits the improvement of bias after bias-adjustment for surface solar irradiance for the 55

sites. Absolute values of the bias after adjustment are coded in three colours: green for absolute value < 5 W m$^{-2}$, yellow for 5<value< 10 W m$^{-2}$, red for value>10 W m$^{-2}$. Change in bias is coded by symbols: circle for changes in absolute value less than 5 W m$^{-2}$, downward triangle for improvement in bias, and an upward triangle for degradation. The size of the triangles increases with increasing absolute values of the bias. For example, a green downward triangle means that the bias has been



decreased (downward triangle, i.e. improvement) and that after bias adjustment, the absolute value of the bias is less than 5 W m$^{-2}$. One may see that there is an improvement or *status quo* for all stations, i.e. there is no upward triangle, only circles and downward triangles. 22 stations out of 57 exhibit a bias less than 5 W m$^{-2}$ in their absolute values, which is a strong improvement compared to the 12 for the original ERA-Interim data.

Once daily means are adjusted, the ratio between the original ERA daily mean and the adjusted daily mean is applied to each of the eight 3-h irradiances within each day. Therefore, no alteration is made to the diurnal cycle of irradiance. This yields the final set of bias-adjusted 3-h surface solar irradiance.

## 5 Discussion

As stated earlier in the paper, the work reported here is specifically targeting energy sector applications; however the bias adjustment carried out here could be applied to a wide range of potential applications. ECEM and its users plan to use both the adjusted and unadjusted ERA-Interim gridded products through ESCIIs (Energy System Climate Impact Indicators), which will relate the climate variables to energy-relevant indices. Whether the bias adjustments improve agreement between these ESCIIs and the direct measures of energy production (e.g. renewable energy from solar and wind farms) is a simple way of

assessing their effectiveness.

The WATCH bias-adjusted datasets developed by Weedon et al. (2011, 2014) have been used extensively, based on citation counts, but they cover a much larger region than our European window. Our dataset applies adjustments to the distributions of a similar set of variables, providing daily and 6-hourly estimates.  Outside the Energy sector, the bias-adjusted datasets could

be used for driving hydrological and land-surface models in a similar way to Orth and Seneviratne (2015). Our bias adjustments, therefore, could be assessed beyond the Energy sector. For Europe, they could be compared with WFDEI data (often referred to as forcing data in hydrology, as opposed to bias-adjusted Reanalyses) through comparison of results from hydrologic and/or crop climate models (e.g. using discharge or yield data). Bias adjustment ought to be an improvement, in a similar way to the assessments we will make with ESCIIs.

Is there a way of simultaneously bias adjusting all variables, or at least in pairs to start with? Whereas Weedon et al. (2014) have not attempted multivariate adjustment, this is being tested in our project. However, as the number of variables increases this becomes more impractical. If a universal method could be found, the usefulness of the approach can be assessed through ESCIIs and discharge/yield data (i.e. using variables external to Reanalysis) which we would expect to be best simulated just

as if we had perfect observational data. Within ECEM, we have experimented with multivariate bias adjustment (using wind speed and temperature), but the results are dependent on the availability of adequate station data for variables measured together (Parie, Pers. Comm.). Access to data is a crucial aspect of all the datasets used in this study. ERA-Interim would be



improved with greater numbers of station input data, as would E-OBS and the other data products considered in this paper. Improved access, however, is unlikely to reduce the need for bias adjustment.

### 6 The ECEM dataset: description and how to access it

All the ERA-Interim (original and bias-adjusted) are available as netcdf files from the Climate Data Store (CDS) of the Copernicus Climate Data Service. As this CDS is currently being developed, this ftp site (ftp://ecem.climate.copernicus.eu) can be currently used to access all files discussed in this paper. This site currently has no password, but once on the CDS, there will likely be a registration procedure. Datasets are named according to ECEM project. The original or unadjusted filenames have 'noc' in the file name. They are, for air temperature (T2M), dewpoint temperature (DP), solar irradiance (SSR), wind
speed (WS) and precipitation (TP)

H_ERI_ECMW_T159_T2M_0002m_EUR1_22E27N_45W72N_050d_IN_TIM_19790101_20141231_06h_NA_noc_org_NA_NAA.nc

H_ERI_ECMW_T159_DP__0002m_EUR1_22E27N_45W72N_050d_IN_TIM_19790101_20141231_06h_NA_noc_org_NA_NAA.nc

H_ERI_ECMW_T159_SSR_0000m_EUR1_22E27N_45W72N_050d_IN_TIM_19790101_20141231_03h_NA_noc_org_NA_NAA.nc

H_ERI_ECMW_T159_WS__0010m_EUR1_22E27N_45W72N_050d_IN_TIM_19790101_20141231_06h_NA_noc_org_NA_NAA.nc

H_ERI_ECMW_T159_TP__0000m_EUR1_22E27N_45W72N_050d_IN_TIM_19790101_20141231_01d_NA_noc_org_NA_NAA.nc

A_NAA.nc

The adjusted files are labelled similarly, but include the distribution and 'bc' instead of 'noc'. So for air temperature and dewpoint they include 'nbc', for solar irradiance 'qbc', for wind speed 'wbc' and precipitation 'gbc'. A final file contains the bias adjusted relative humidity file.

H_ERI_ECMW_T159_T2M_0002m_EUR1_22E27N_45W72N_050d_IN_TIM_19790101_20141231_06h_NA_nbc_org_NA_NAA.nc

H_ERI_ECMW_T159_DP__0002m_EUR1_22E27N_45W72N_050d_IN_TIM_19790101_20141231_06h_NA_nbc_org_NA_NAA.nc

H_ERI_ECMW_T159_SSR_0000m_EUR1_22E27N_45W72N_050d_IN_TIM_19790101_20141231_03h_NA_qbc_org_NA_NAA.nc

H_ERI_ECMW_T159_WS__0010m_EUR1_22E27N_45W72N_050d_IN_TIM_19790101_20141231_06h_NA_wbc_org_NA_NAA.nc



H_ERI_ECMW_T159_TP__0000m_EUR1_22E27N_45W72N_050d_IN_TIM_19790101_20141231_01d_NA_gbc_org_N
A_NAA.nc

H_ERI_ECMW_T159_RH__0002m_EUR1_22E27N_45W72N_050d_DR_TIM_19790101_20141231_06h_NA_nbc_org_
NA_NAA.nc

Two example locations for each variable of the distributional comparisons are given in the paper (Figures 2, 3, 6, 7, 10, 11, 14
and 15). The ftp site also includes all the distributional comparisons as pdfs, with the stations order by their WMO number,
when comparing with HadISD, and by latitude then longitude when comparing with E-OBS. These files have these name, for
air temperature (Tmean), dewpoint temperature, wind speed (ws) and precipitation (dly_precip), respectively

adjERA_and_ERA_vs_EOBS_dly_Tmean_PDFs_1981-2010.pdf

adjERA_and_ERA_vs_HadISD_dly_dewpoint_PDFs_1979-2014.pdf

adjERA_and_ERA_vs_HadISD_ws_PDFs_1979-2014.pdf

ERA_and_adjERA_vs_EOBS_dly_precip_PDFs_1981-2010_1p0.pdf

**7 Sources of data used**

ERA Interim data were downloaded from here (http://apps.ecmwf.int/datasets/data/interim-full-daily/levtype=sfc/) and our
regridded version at the 0.5° by 0.5° grid is available as the original dataset (see Section 6)

E-OBS for both daily air temperature and precipitation grids (http://www.ecad.eu/download/ensembles/ensembles.php)

CRU for both monthly air temperature and precipitation grids (CRU TS 3.23, https://crudata.uea.ac.uk/cru/data/hrg/)

GPCC for monthly precipitation grids (https://www.dwd.de/EN/ourservices/gpcc/gpcc.html).

HadISD for sub-daily station data for wind speeds and dewpoint temperatures (http://www.metoffice.gov.uk/hadobs/hadisd/).
For all the above datasets, the data are freely available for use, but this is qualified on some sites as use is for research and
educational purposes and it may be necessary to register to gain access.

Station data for surface solar irradiance were downloaded from the web site (www.wrdc.mgo.rssi.ru ) of the World Radiation

Data Center (WRDC) after registration. Data are available only for research and educational communities of the countries
participating to WMO for non-commercial activities.

HelioClim-3v5 datasets were downloaded from the SoDa Service web site (www.soda-pro.com) managed by the company
Transvalor. Data are available to anyone for free for years 2004-2006 as a GEOSS Data-CORE (GEOSS Data Collection of
Open Resources for Everyone) and for-pay for the most recent years with charge depending on requests and requester.



**Acknowledgements**

The authors would like to acknowledge funding for the European Climatic Energy Mixes (ECEM) project by the Copernicus Climate Change Service, a programme being implemented by the European Centre for Medium-Range Weather Forecasts (ECMWF) on behalf of the European Commission. The specific grant number is 2015/C3S_441_Lot2_UEA. The authors also thank Robert Dunn from the UK MetOffice Hadley Center who kindly extracted the wind speed and wind direction for all stations from the HadISD data set. The authors thank all ground station operators of the WMO network for their valuable measurements. They additionally thank the World Radiation Data Centre for hosting a web site for downloading data. The authors thank the French company Transvalor which is taking care of the SoDa Service for the common good, therefore permitting an efficient access to the HelioClim databases.

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

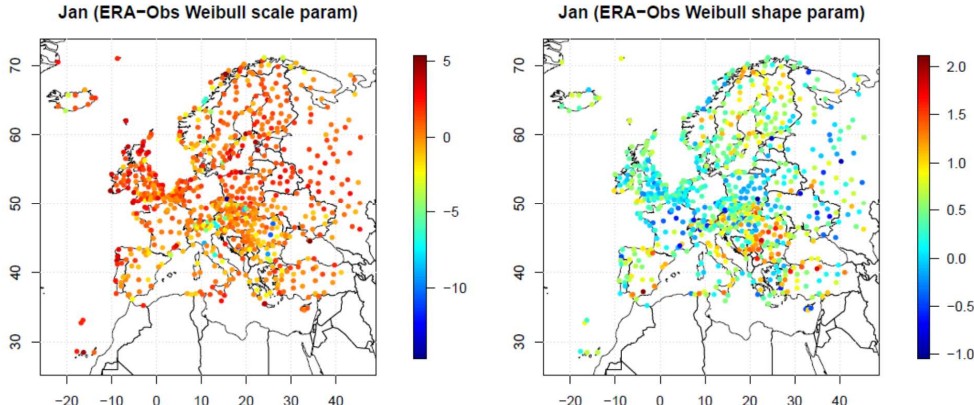

**Figure 1:** Differences in scale and shape parameters of the Weibull distribution between ERA-Interim and HadIDS station observations for wind speed at 10 m. Based on all 6 hourly data for January for 1981-2010.



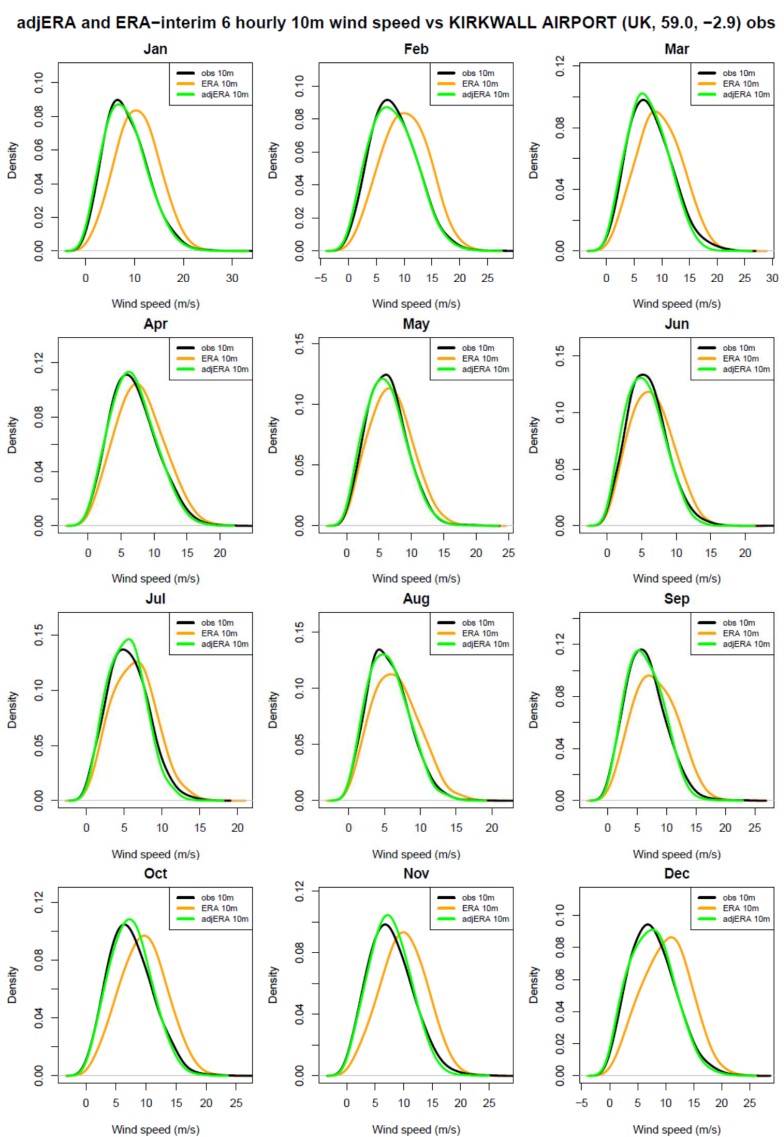

Figure 2: Comparison of statistical distributions of wind speed at 10 m for Kirkwall, Scotland, for observations (black), ERA-Interim (orange) and bias-adjusted ERA-Interim (green), based on all 6 hourly data for the 1981-2010 period.

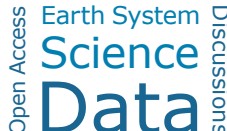



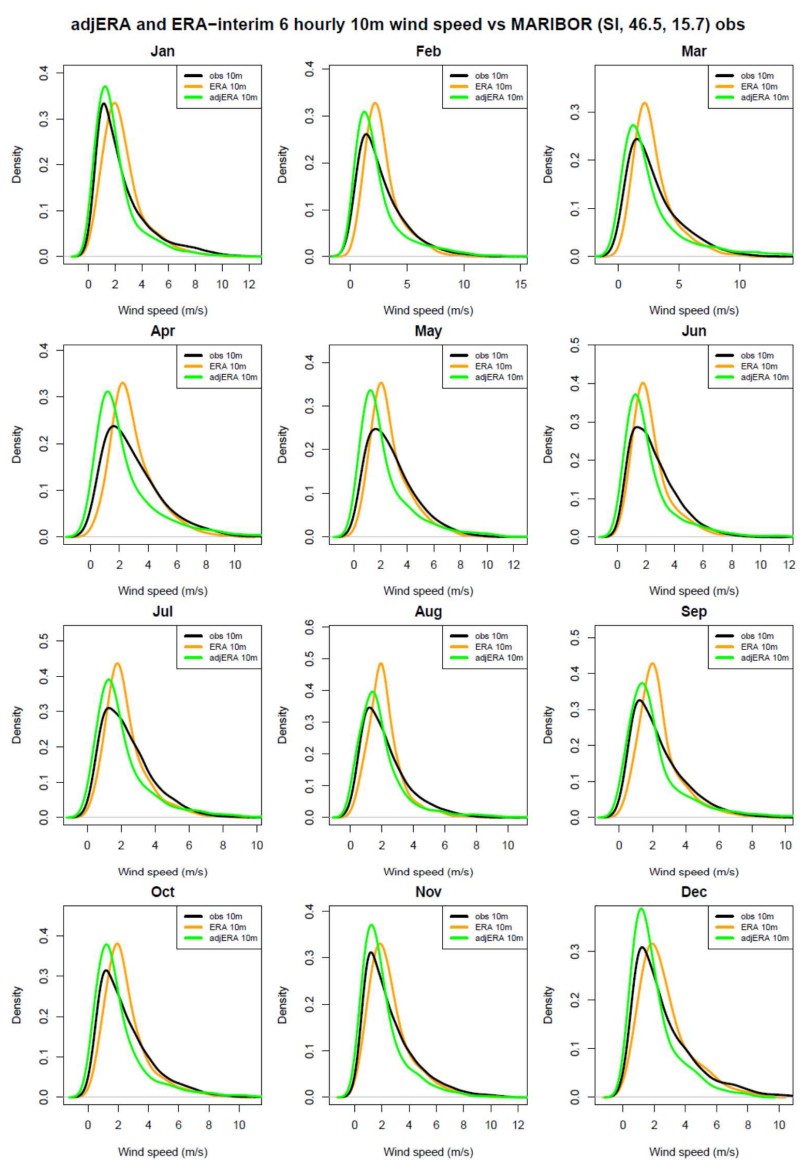

**Figure 3: Comparison of statistical distributions of wind speed at 10 m for Maribor, Slovenia, for observations (black), ERA-Interim (orange) and bias-adjusted ERA-Interim (green), based on all 6 hourly data for the 1981-2010 period.**



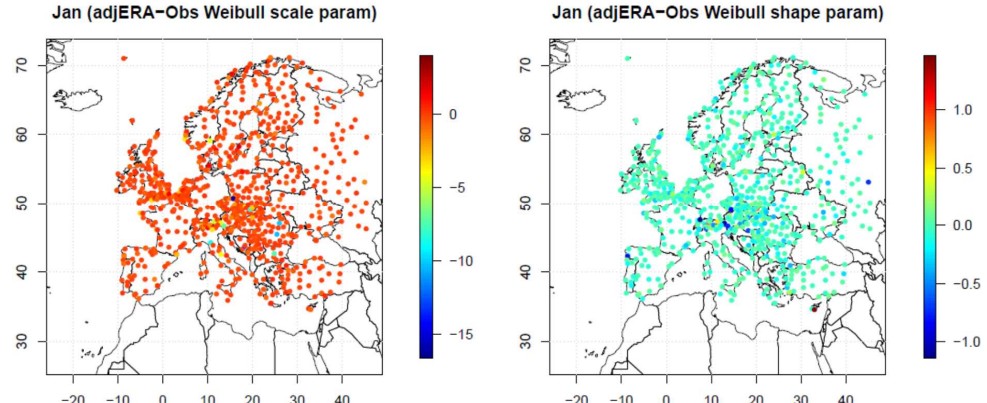

**Figure 4: Differences in scale and shape parameters of the Weibull distribution between bias-adjusted ERA-Interim and HadIDS station observations for wind speed at 10 m. Based on all 6 hourly data for January for 1981-2010.**

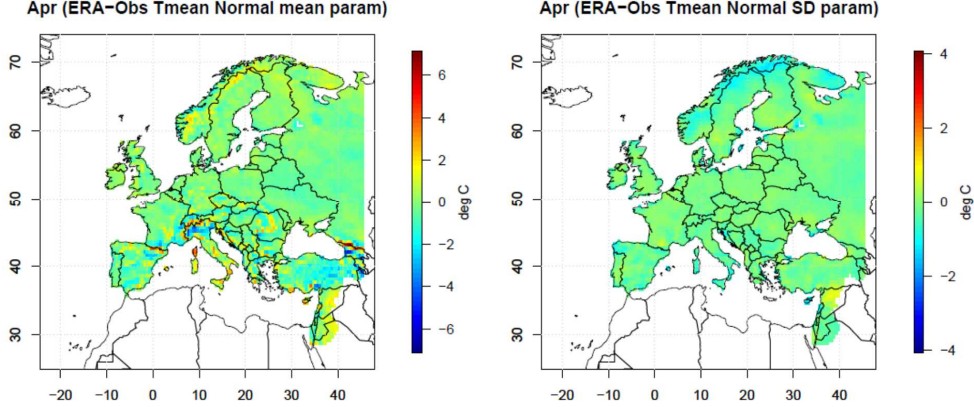

**Figure 5: Differences in means and standard deviations (SD) between ERA-Interim and E-OBS for mean surface air temperature (Tmean). Based on daily data for April for 1981-2010.**





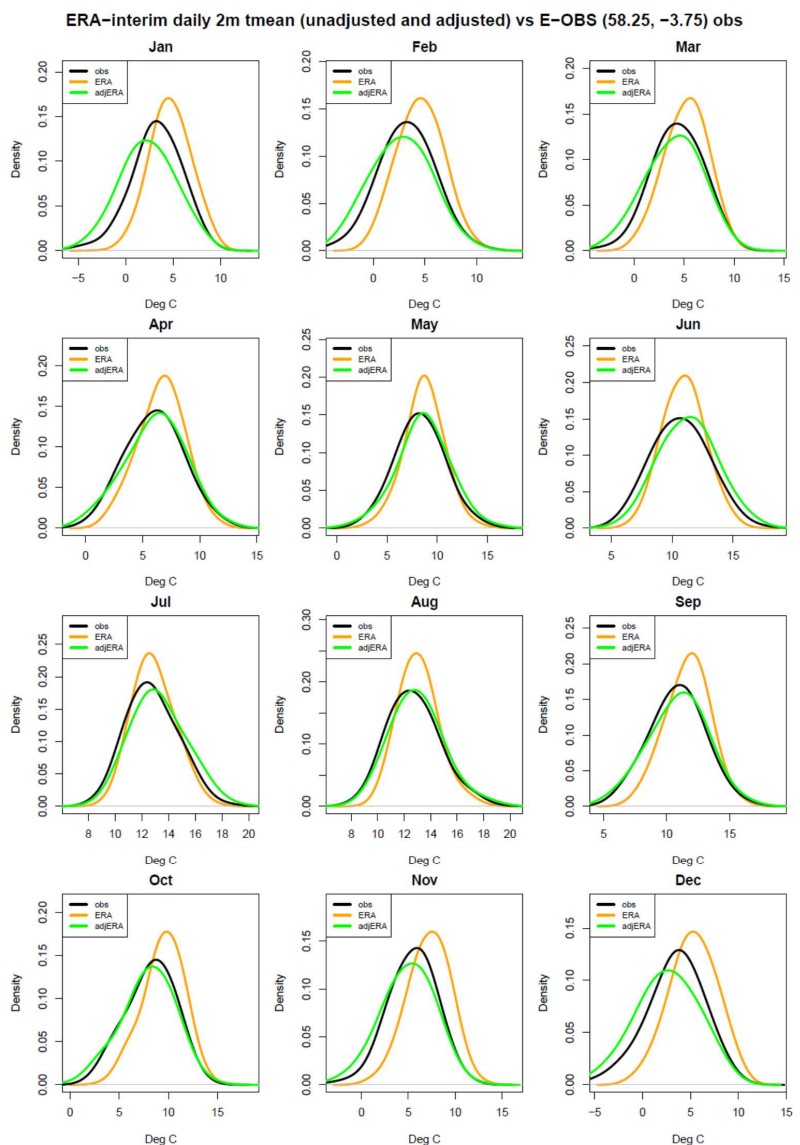

Figure 6: Comparison of statistical distributions of surface air temperature for Northern Scotland (lat: 58.25°N, lon: 3.75°W), for observations (black), ERA-Interim (orange) and bias-adjusted ERA-Interim (green), based on daily data for the 1981-2010 period.


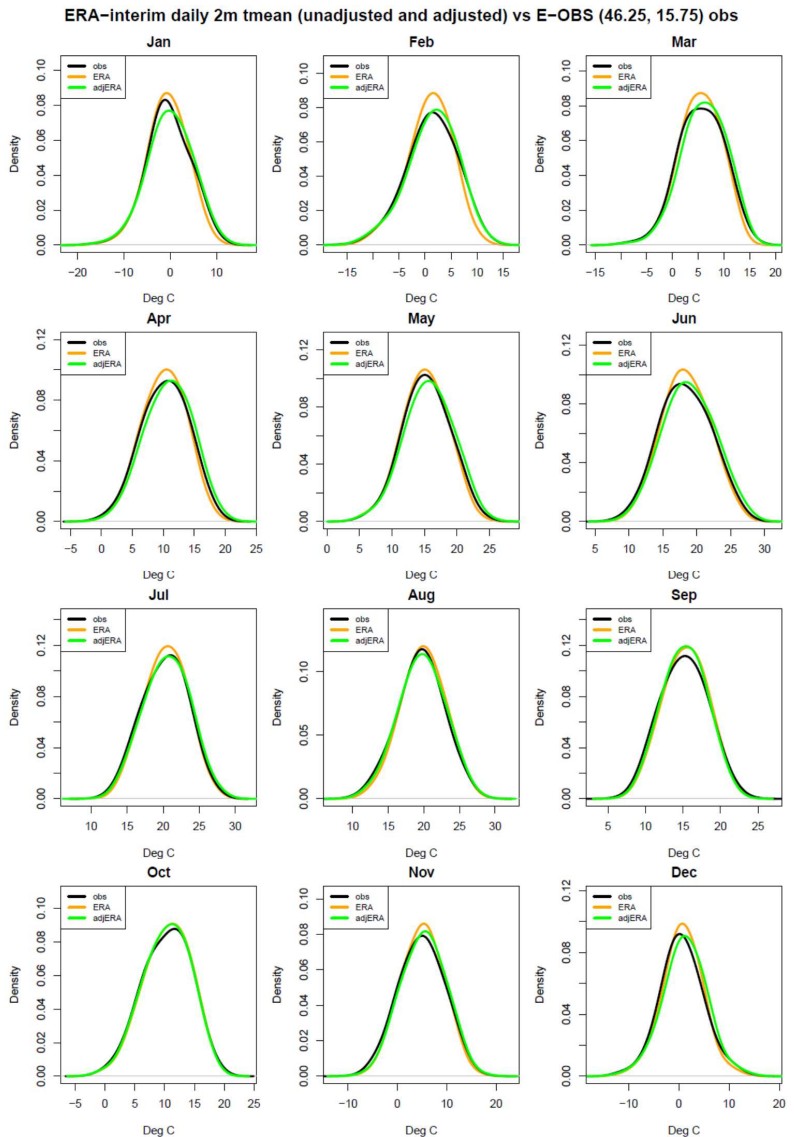

**Figure 7:** Comparison of statistical distributions of surface air temperature for Slovenia (lat: 46.25°N lon: 15.75°E), for observations (black), ERA-Interim (orange) and bias-adjusted ERA-Interim (green), based on daily data for the 1981-2010 period.



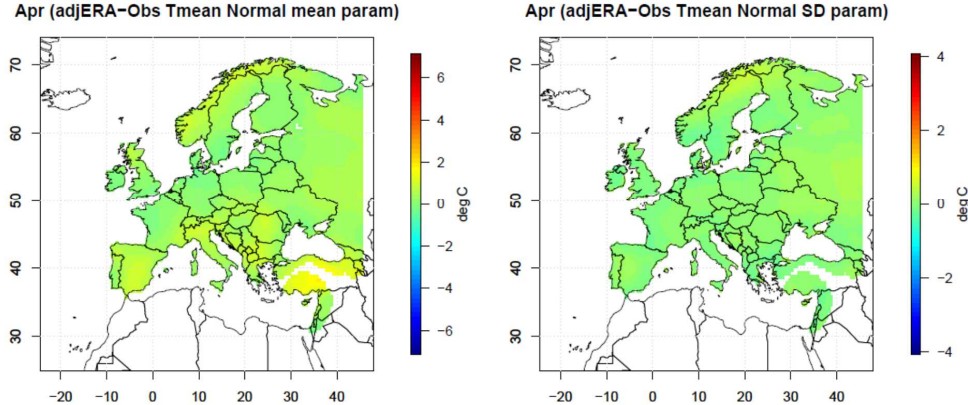

**Figure 8: Differences in means and standard deviations (SD) between bias-adjusted ERA-Interim and E-OBS for mean surface air temperature (Tmean). Based on all data for April for 1981-2010.**

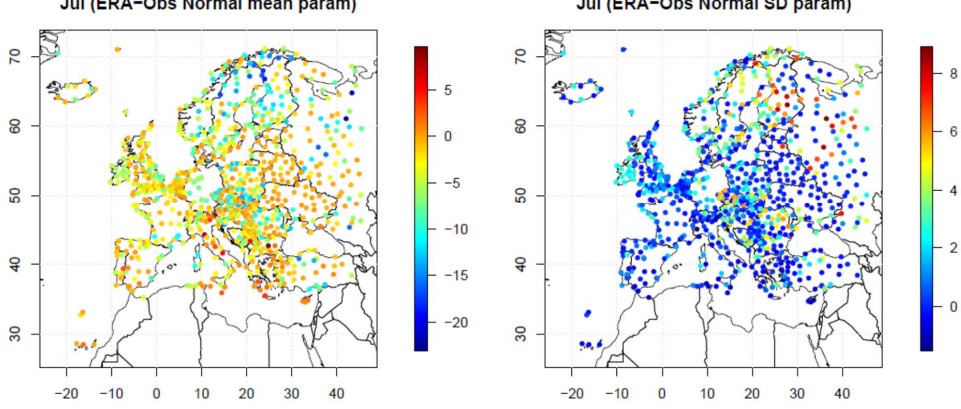

**Figure 9: Differences in means and standard deviations (SD) between ERA-Interim and HadISD for dewpoint temperature (°C). Based on daily data for July for 1981-2010.**



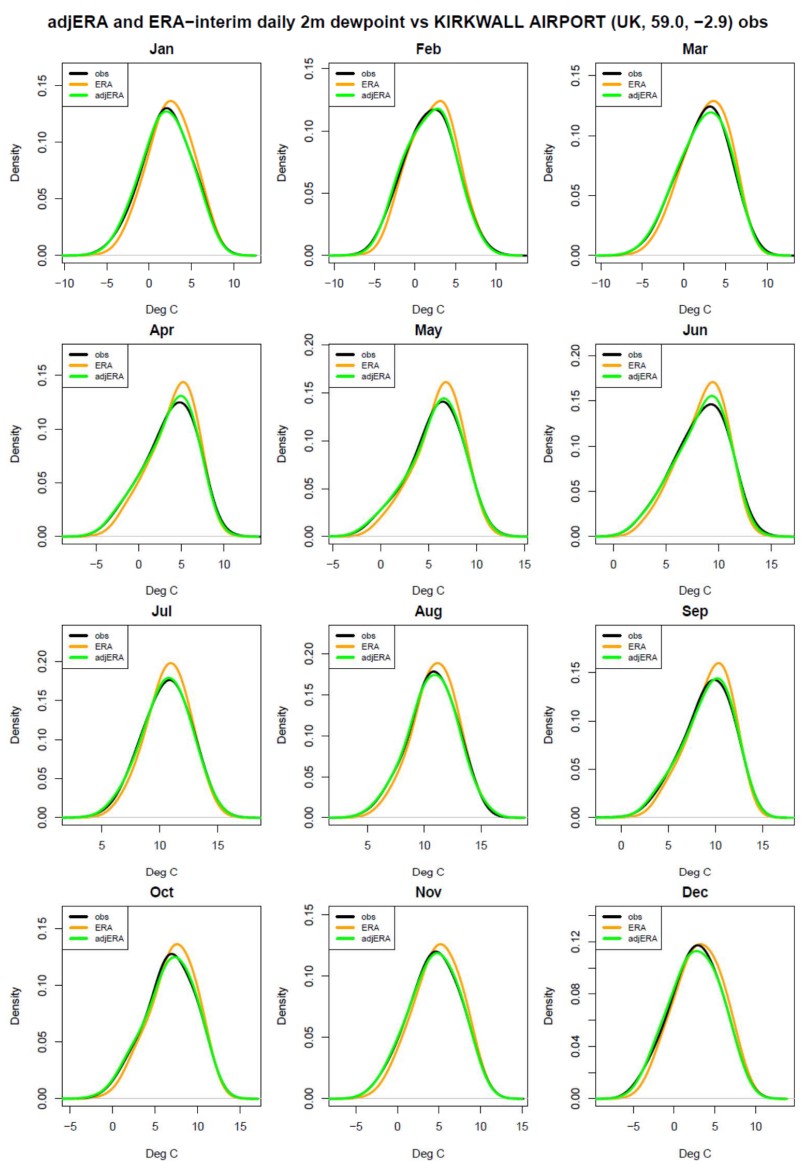

**Figure 10: Comparison of statistical distributions of dewpoint temperature for Kirkwall, for HadISD observations (black), ERA-Interim (orange) and bias-adjusted ERA-Interim (green), based on daily data for the 1981-2010 period.**



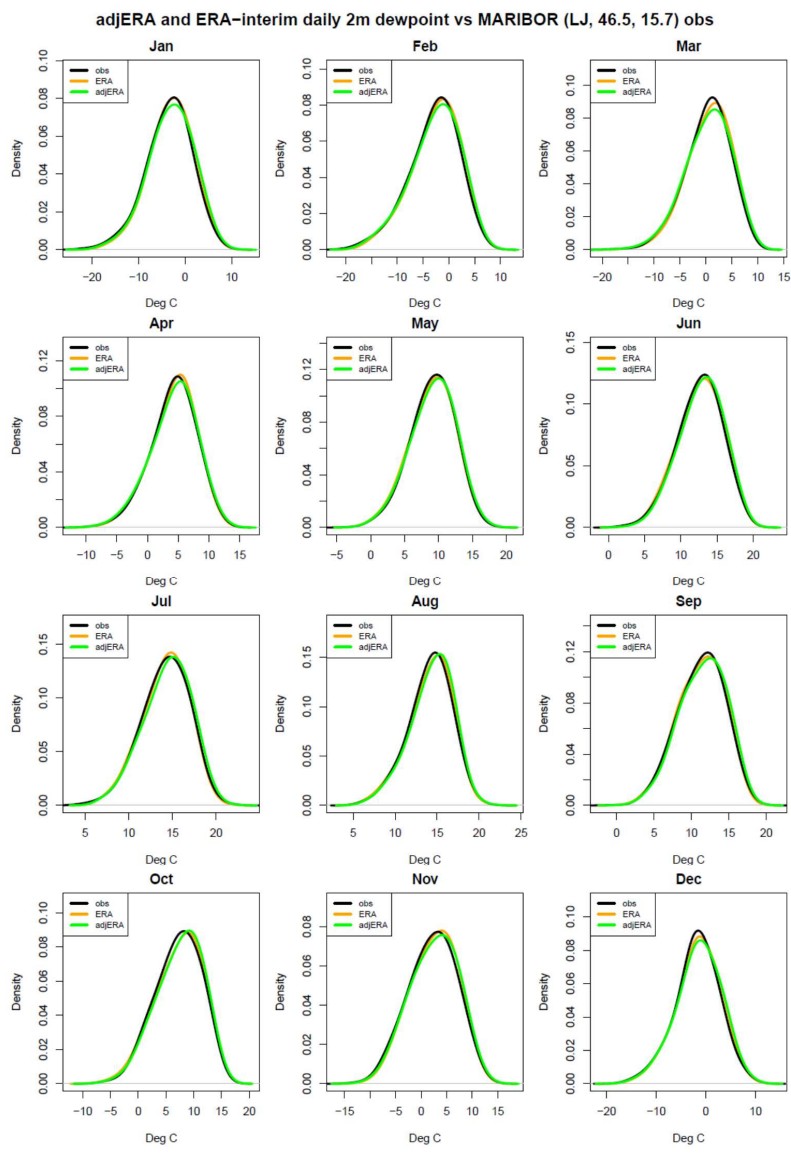

**Figure 11: Comparison of statistical distributions of dewpoint temperature for Maribor, for HadISD observations (black), ERA-Interim (orange) and bias-adjusted ERA-Interim (green), based on daily data for the 1981-2010 period.**



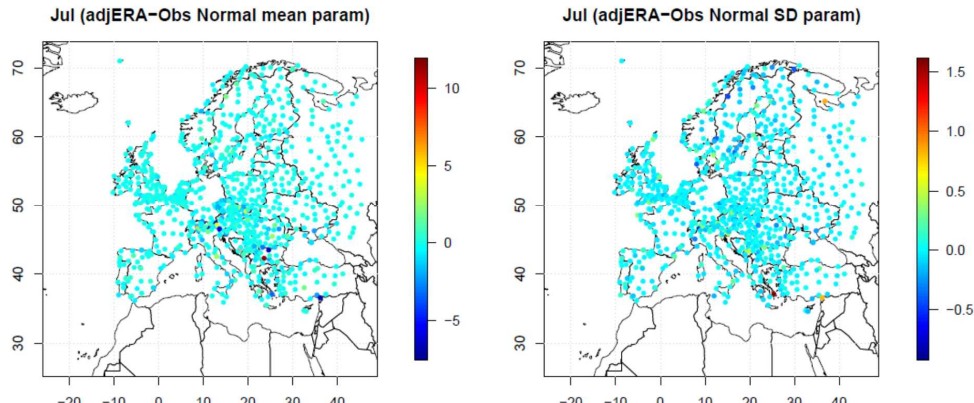

**Figure 12: Differences in means and standard deviations (SD) between bias-adjusted ERA-Interim and HadISD for dewpoint temperature (°C). Based on daily data for July for 1981-2010.**

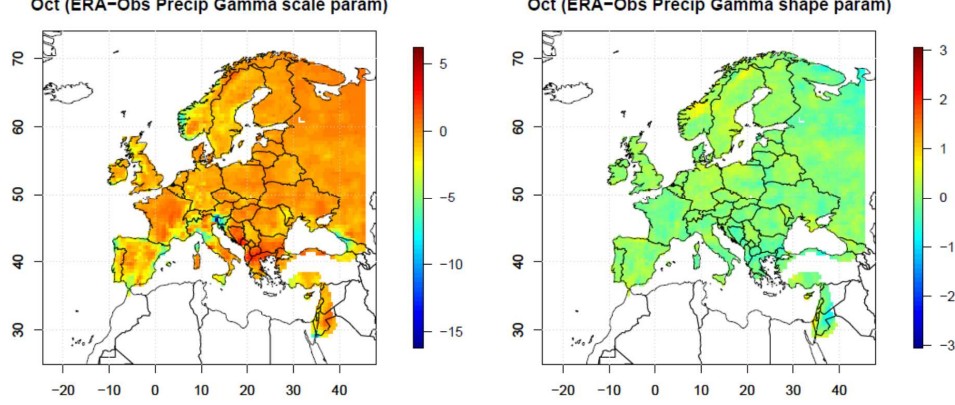

**Figure 13: Differences in scale and shape parameters of the gamma distribution between ERA-Interim and E-OBS for precipitation daily totals > 1 mm. Based on daily precipitation totals for October for 1981-2010.**



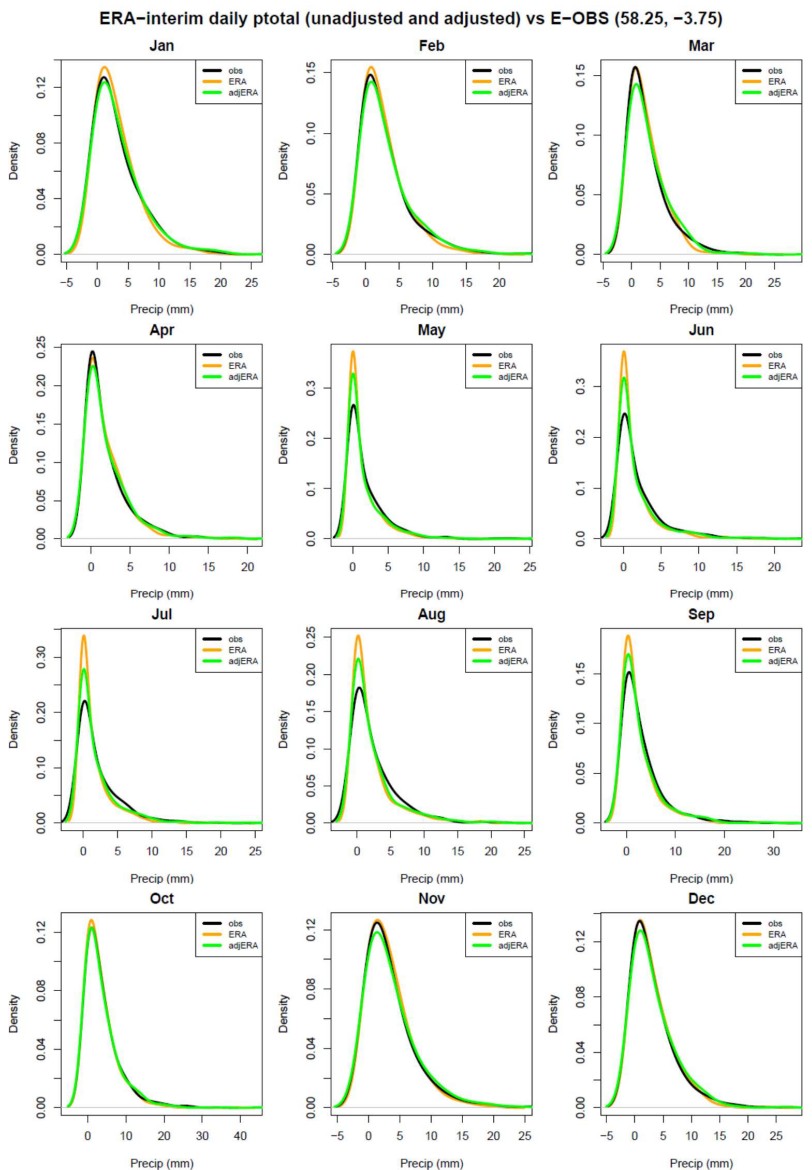

**Figure 14: Comparison of statistical distributions of precipitation daily totals for Northern Scotland (lat: 58.25°N, lon: -3.75°W), for observations (black), ERA-Interim (orange) and bias-adjusted ERA-Interim (green), based on the 1981-2010 period.**



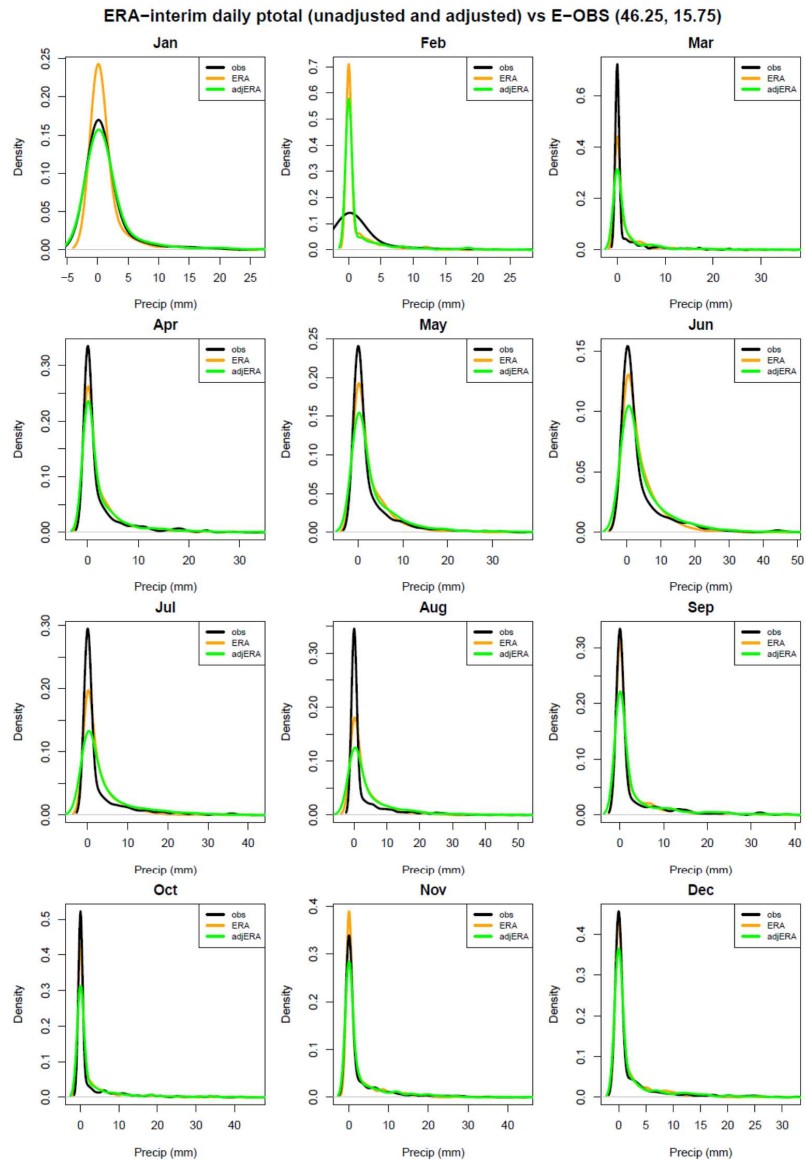

**Figure 15: Comparison of statistical distributions of precipitation daily totals for Slovenia (lat: 46.25°N, lon: 15.75°E), for observations (black), ERA-Interim (orange) and bias-adjusted ERA-Interim (green), based on the 1981-2010 period.**




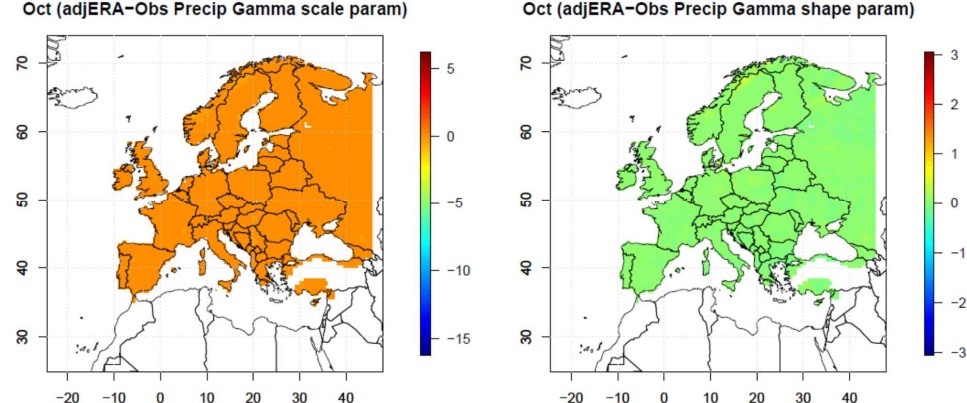

**Figure 16: Differences in scale and shape parameters of the gamma distribution between bias-adjusted ERA-Interim and E-OBS**
5 **for precipitation daily totals > 1 mm. Based on daily precipitation totals for October for 1981-2010.**



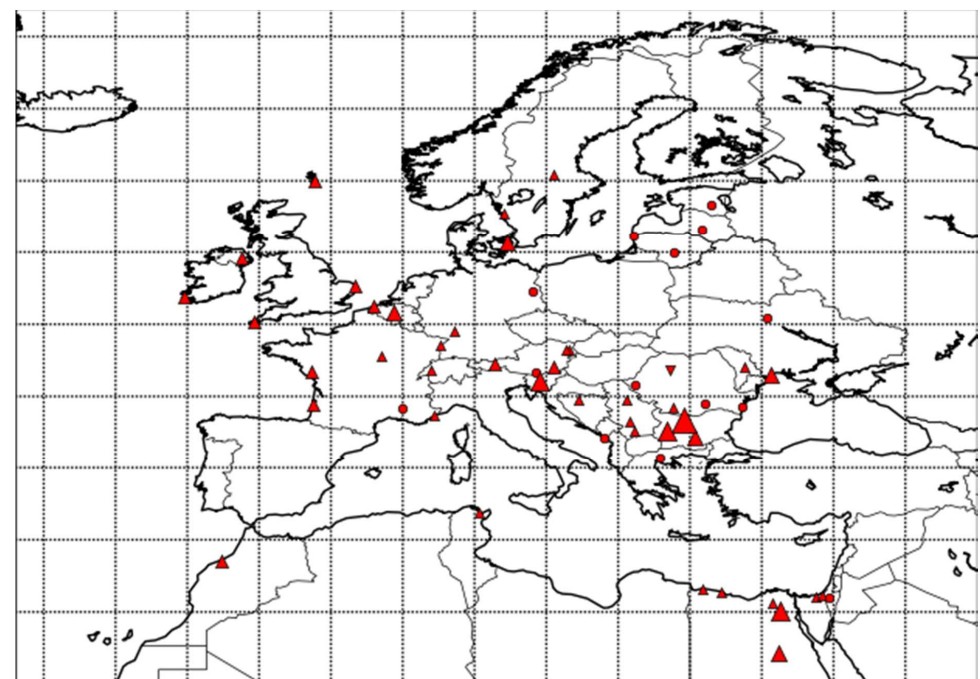

**Figure 17: Bias for ERA-Interim vs ground observations of daily mean of surface solar irradiance for 57 stations. Triangles downward mean a negative bias less than -5 W m⁻², triangles upward mean a positive bias greater than 5 W m⁻² and circles mean an absolute value of the bias less than 5 W m⁻². The size of the triangles increases with increasing absolute value of the bias.**



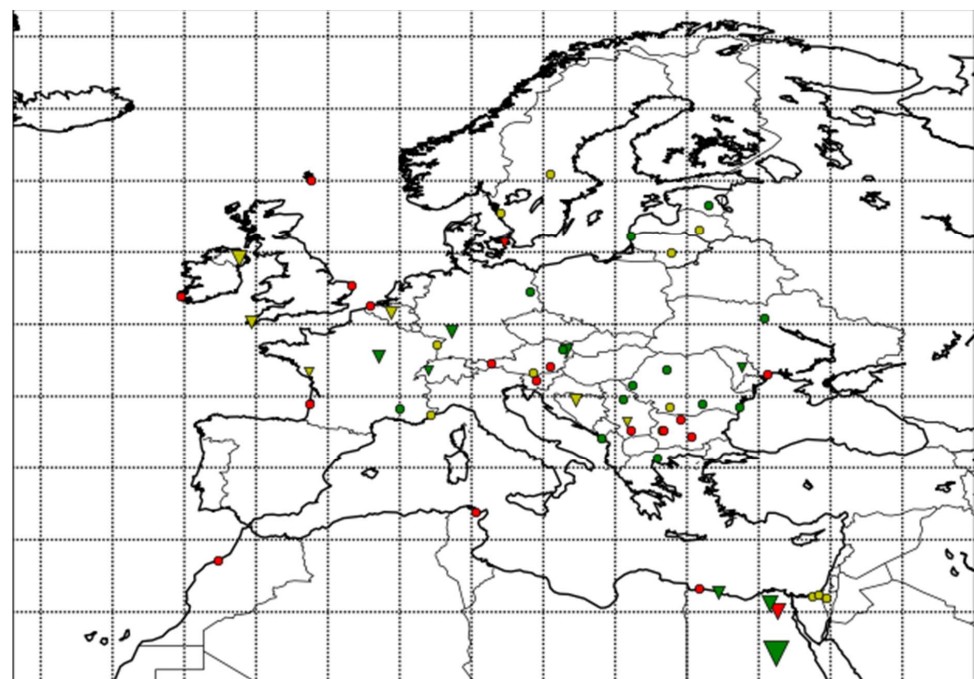

**Figure 18: Improvement of bias after bias-adjustment for daily mean of surface solar irradiance. Absolute values of the bias after adjustment are coded in three colours: green for absolute value < 5 W m⁻², yellow for 5<value< 10 W m⁻², red for value>10 W m⁻². Change in bias is coded by symbols: circle for changes in absolute value less than 5 W m⁻², triangle downward for improvement in**
5   **bias, and triangle upward for degradation. The size of the triangles increases with increasing absolute value of the bias.**