# Peer review of "Using ERA-Interim Reanalysis output for creating datasets of energyrelevant climate variables"

_Earth System Science Data, 2016_

## Referee Comment (RC1) · G.P. Weedon (Referee) · 8 Feb 2017

General comments.

Jones et al present a dataset of meteorological variables at half-degree resolution for Europe based on bias correction of ERA-Interim surface reanalysis data. They use observations to correct the distributions (e.g. mean and skewness) of the variables rather than just correcting the means. The new product is designed to be of use for the energy sector.

Although well presented and clear, there are two overriding problems with the manuscript in its current form. Firstly, their methodologies have limitations that are

not explored in appropriate depth - particularly with regard to their revised distributions for near-surface air temperature, relative humidity and for daily precipitation totals. Secondly, although the text repeatedly refers to an earlier dataset, also at half-degree resolution and based on bias correction of ERA-Interim reanalysis data (i.e. the WFDEI, Weedon et al. 2014 WRR) there is no comparison made between the new data and the existing dataset or other existing alternatives. Since they have corrected both the means and distributions of the ERA-Interim data I was surprised that there is no demonstration that compared to observations the new dataset represents an improvement over the existing datasets. This would both demonstrate that it is more useful for the energy sector and support their claim that their methodology is worth applying in other circumstances.

Specific comments.

a) Dew point temperature.

The authors start the manuscript by stating that previous datasets applied bias corrections that were independent for each meteorological variable. This is misleading since the WATCH Forcing Data methodology (WFD), that was used to create the WFDEI (i.e. Weedon et al 2010 Tech Rep cited by Weedon et al 2011 JHM, Weedon et al 2014 WRR) specifically adopted the approach specified by Cosgrove et al 2003 (JGR). The Cosgrove et al approach was also used by another reanalysis-based dataset not mentioned: the Princeton Global Forcing (PGF) of Sheffield et al 2006 (J Clim).

The Cosgrove et al approach ensures that changes to near-surface temperature are incorporated into changes in air pressure, specific humidity and then downwards long-wave radiation by using sequential processing (maintaining the original covariances). Surprisingly, despite raising this issue Jones et al adjusted air temperatures independently of dew point temperatures and consequently, as discussed on manuscript page 9, encountered implied relative humidities exceeding 100% (specifically avoided in the WFD, WFDEI and PGF). I strongly recommend that the authors design an approach

that ensures that adjustments to air temperature and dew point temperature are applied in a consistent fashion - to avoid the possibility of their implied relative humidity exceeding 100%.

b) Air temperature.

The Cosgrove et al (2003) methodology allows for elevation differences between observation sites and the mean elevation of reanalysis grid boxes. There are elevation differences between the heights of the EOBS temperature sites and the half degree elevations of the adjusted products. Hence an important step is to ensure that firstly the EOBS temperature data have been adjusted, using a lapse rate, to sea level, then interpolated to half degree and then recover the interpolated EOBS temperatures at the height of the ERA-Interim half degree grid boxes. As such the resulting interpolated EOBS data can then be used to correct the interpolated ERA Interim temperatures (similarly interpolated at sea level, Cosgrove et al 2003). This is such a critical processing step that I assume it was an oversight that it was not stated by the authors.

The average bias of their corrected air temperature product is shown for April in Figure 8. There are numerous areas within the map where the mean air temperatures do not match the observations to within a small part of a degree Celcius especially in mountainous areas and Turkey (where there is a bias of more than 2 oC). Since their methodology is designed to fix the means as part of the bias correction I find such differences to be surprising at the least and it definitely requires explicit explanation in the text. The text should also describe how and why the air temperature bias maps vary by month or season.

Furthermore, the distributions of corrected air temperatures shown for the Scottish site differ substantially in terms of the most probable value by around one degree in winter (Dec and Jan). Somehow the modal temperature has been changed from more than the EOBS mode in the original ERA-Interim data to less than the EOBS mode. I would have expected the modes to align everywhere after the adjustments. Again

this requires specific comment. It is conceivable that the means do in fact match with say 0.1oC whereas the modes differ substantially for December and January, but this needs explanation/clarification for the reader. Are there similar issues elsewhere and if so why?

c) Precipitation.

Aside from consideration of numbers of wet days, a gamma distribution has been used to correct the distribution of daily precipitation totals. Unfortunately, the illustrated results are not encouraging. For the Scottish example in Figure 14, in May to September the distribution is improved compared to the original data, but there appears to be room for further improvement. Nevertheless, illustration of the WFDEI and/or PGF results would probably put the adjusted Scottish values into a better perspective.

In the case from Slovenia (Figure 15) the results suggest significant methodological problems. In April to August it is clear that the adjustments have made the distributions worse than the original ERA Interim data. In these months not only is the modal value further from the observations than the raw data (green line further from black obs than the orange line), but also at intermediate precipitation totals (2 to 8 mm/day) the corrected distribution is worse than the original data (in this region of the graphs higher). These problems should be discussed in the manuscript.

It is clear that the authors appreciate that different localities require different approaches (manuscript page 10 line 3). Why have the authors not designed code that, when the distribution is not improved by the adjustments the results are reverted to uncorrected values? Piani et al (2010 J Clim - NB not the Piani et al 2010 paper cited by Jones et al) adopted a variety of transfer functions for daily precipitation totals chosen according to their appropriateness by location (ranging from simple to reasonably complex). The figure showing the spatial distribution of the adjusted precipitation results is provided for October (Figure 16). However, we are not told whether other months have worse performance (as is implied for the Slovenia example at least for April to August).

The manuscript definitely needs to appraise the success of the precipitation bias corrections more critically with reference to issues linked to both location and month or season.

d) Comparison to previous datasets.

I am very surprised the authors did not explicitly demonstrate that their data provide advantages in terms of reliability compared to the existing, widely-used datasets that are currently available for the same time interval and the same spatial resolution (i.e. WFDEI and PGF). Both diagrams illustrating the point and a text discussion of the advantages of the new product merit a new section.

Technical corrections.

p2 line 22 The WFDEI now extends to the end of 2014 (not 2013 as stated).

Figure 14 There is no such thing as negative mm/day in precipitation. Regardless of the calculation of the gamma distribution extending to negative values, the plots should be truncated at 0.0 mm/day.

Figures 17 and 18: a) Remove the grid lines (they are confusing and were not used on the other maps). b) Simplify the information by using a single symbol coloured according to the size of bias (this makes the maps more comparable to the previous way biases were shown). c) Illustrate a colour scale for easy determination of the size of the bias. d) Do not use green and red at the extremes of a distribution as it disadvantages colour blind readers. Simply use blue at the opposite extreme to red.

Recommendation:

Major revisions addressing the concerns listed above.

graham.weedon@metoffice.gov.uk, 8th Feb 2017.
* * *

---

## Referee Comment (RC2) · H. F. Goessling (Referee) · 22 Feb 2017

**General comments**

(Please note that I have not read *RC1* before submitting my review to maintain independence.)

The authors describe how they obtained a bias-adjusted dataset for energy-sector relevant parameters from the ERA-Interim reanalysis. I have a substantial number of remarks. Overall, I am not really convinced that the bias-adjustment as conducted here is a significant step forward regarding the quality of the data compared to the unadjusted data. However, I certainly think that the data set deserves being published

and documented through this paper – if certain points are clarified and the quality of the results are discussed more extensively and critically. This pertains in particular to the required independence between the data used for bias adjustment and the data used to assess the pre- and post-adjustment quality. I therefore recommend to accept the paper subject to major revisions.

My main points of criticism include:

1. If I am not mistaken, only in one case – for surface solar irradiance – an independent data set is used to assess the effects of the bias-adjustment in a conclusive way. In all other cases it appears that the data used for the adjustment are then used to assess the post-adjustment "bias" – which is not exactly zero only because of certain interpolation/remapping steps. If I am wrong here, please explain your approach in a way that prevents other readers from repeating my misunderstanding.

2. It is also unclear to what extent the station and derived gridded data are independent of data that have entered the assimilation system used to generate the reanalysis in the first place, and what the impact of interdependence might be (e.g., for near-surface temperature).

3. It is stated that the figures showing the statistical data distributions (Figs. 2, 3, 6, 7, ...) *exhibit the [corresponding] distribution fits.* I think that this is not correct, but that the figures show the empirical data densities obtained by some smoothing, e.g., with a Gaussian filter. As a consequence, also those parameters that are positive by definition (wind speed and precipitation) exhibit non-zero densities at negative values (where the theoretical distribution fits should have zero density). This should be corrected in the text and also explained in the figure captions. Also, I suggest to add the two corresponding theoretical distributions to each of the plot panels (e.g., as dashed curves), so one can judge how well the empirical distributions are approximated by the theoretical distributions.

[Figure]

**Specific comments**

*Abstract:* I suggest to mention in the abstract that univariate adjustment is applied. (To my understanding, multivariate approaches are mentioned in the paper several times, but were not applied to generate the present data set.)

*P1 L17–18 – The benefit of performing bias-adjustment is demonstrated by comparing initial and bias-adjusted ERA-Interim data against observations:* See general comment 1.

*P1 L22–23 – These are reconstructions of past climates produced through the blending of observations with physical/numerical models which have been developed explicitly for climate monitoring and research:* Speaking of the use of reanalyses for climate monitoring (which in my view implies the consideration of long-term trends), I recommend to add a comment of caution regarding possible spurious trends that can arise from changes in the observing system (e.g., Bengtsson et al. 2004 *Can climate trends be calculated from reanalysis data?*).

*P2 L5 – climate model:* I think this should rather read *weather (forecast) model*, or maybe less specifically *numerical model*.

*P2 L5 – amount of observational data:* Could be complemented to *amount, type, and quality of observational data*.

*P2 L12–13 – the bias may be larger for [...] regions of sparse station coverage:* If I understand correctly, this statement holds only if this refers to those stations from which the data enter the assimilation system. In this context a clear distinction should be made between assimilation-related stations and stations used for the bias adjustment. There is probably strong overlap, but I doubt that the two sets of stations are identical (not speaking of the various other data types that enter the assimilation system).

*P2 L31–32 – The benefit of performing bias-adjustment is demonstrated by comparing initial and bias-adjusted data against station observations and gridded observation*

*products:* See the general comment regarding independence of the data used to assess the bias-adjusted data.

*Section 2:* I would find it very helpful if a table that gives an overview over all the used data sets could be provided, stating if they are station data or gridded (and on what grids, originally and after interpolation), what their time resolution and period is, what they are used for in this paper (bias-adjustment versus independent assessment versus ...), and so forth.

*Section 2:* I think the WRDC data should also be introduced in this section.

*P3 L8–9 – This section provides details of ERA-Interim, and the various gridded and station observation datasets used to assess the quality of this Reanalysis:* The various observation datasets are also the ones used for the bias-adjustment, right? That should be clarified.

*P3 L9–10 – may be regridded* and *can be interpolated:* It should be described only what has actually been done for this paper and associated data set, not what "may be done".

*P3 L14 – 3 h (forecast):* Well, in this case every other time step is still an analysis, right?

*P3 L1–5:* I don't find where CRU and GPCC are actually used in the remainder of the paper, except where it's stated that the authors "began by comparing ERA-Interim against the gridded observational products at the monthly timescale", followed by some statements that are, however, not supported by figures. It is stated that "they are of potential use" in certain circumstances, but it appears that they do not enter the associated data set. If that's right, it should not be stated that these products are used (as done in *P3 L24*).

*Section 2.4 – HelioClim:* I am wondering whether this satellite-derived data set might have been adjusted to the WRDC data – in which case the improvements shown in

Fig. 8 might be not so conclusive? Please clarify.

*Section 3:* Given that, to my understanding, only univariate adjustment is applied to generate the present datasets, I find it distracting that much of this section is about methods for multivariate adjustment.

*P5 L8 – Bias adjustment and bias correction are widely-used terms for the assessment of climate model output:* One can assess climate model output without having bias adjustment in mind.

*P5 L11–12 – the assumption is made there that Reanalyses are correct:* I don't think that anyone assumes absolute correctness of reanalysis data – which can equally not be said of station data.

*P6 L26 – for each month:* I guess this is meant in a climatological sense, right? So I recommend to modify this to something like *for each month of the year*.

*P7 L4–6 – The similarity of the two distributions in terms of their scale and shape parameters indicates that bias adjustment could be achieved by replacing the ERA-Interim scale and shape parameters with those inferred from the HadISD stations:* Im not convinced by this argument. If a transformation based on a certain distribution type is "valid" should in my view rather depend on the degree to which the two samples are consistent with the theoretical distribution used. The fact that the parameters are similar rather indicates that the bias is not very large, after all.

*P7 L12–13 – Figures 2 and 3 exhibit the Weibull distribution fits of the HadISD obser-vations:* I think that this is not correct, see my corresponding general comment.

*P7 L26–27 – the number of stations in some parts of Europe is less dense, so involving greater extrapolation from stations more distant from the grid boxes:* I fail to understand why this should result in larger discrepancies in less dense areas. After all, locations with no stations close to them are not evaluated – because there are no stations at that location to do that! Shouldn't it be quite the opposite, namely, that isolated stations

should show only small discrepancies after the procedure because the nearest grid box is influenced almost exclusively by that station, implying very weak interpolation effects? Please clarify. (This point is closely related to my general comment regarding missing independence of data used for adjustment and (the same) data used to check the post-adjustment "bias".)

*P8 L3 – for the nearest E-OBS grid box:* Is E-OBS not on the same grid as the interpolated ERA-Interim data? Please clarify.

*P8 L4 – Data are then normalized as in Equations 4 and 5:* It appears that the normalization is described only be Eq. 4, whereas Eq. 5 describes the back-transformation.

*P9 L8–11 – Any type of bias adjustment procedure will additionally be influenced by the quality of the station observations, [e.g.,] by the large differences in potential height between some observing locations and the average height field used by ERA-Interim:* Indeed, and I think this point – stations at a particular point not necessarily being representative of a "grid cell" – should be mentioned more prominently and generally in the paper (it doesn't hold just for temperature and humidity, but for all considered parameters).

*P9 Equation 6:* I think that the right-hand-side gives the probability density function rather than the cumulative distribution function (which the left-hand-side implies). In addition, it appears that $\beta$ is the scale parameter and $\alpha$ the shape parameter, rather than the other way around as stated in the subsequent sentence.

*P9 L24–25 – all precipitation amounts below the threshold are set to zero, further improving the agreement between E-OBS and ERA-Interim in the number of dry days per month (i.e. days with rainfall less than the 1.0 mm threshold):* This seems illogical: the modified values were already below the threshold, so they were already "dry days" according to the definition.

*P10 L17–18 – The method 'quantile mapping' applied to KT was preferred (and is*

*used here) as it usually brings improvement with no degradation of the bias, while the other methods often degrade the bias in a noticeable way:* What does this imply for the other parameters where no independent validation was conducted, and where only on adjustment method was applied? It appears to me that this point to some extent compromises the validity of the bias-adjustment approach used in this work.

*P10 L21–23:* Half of this paragraph is just a repetition of the corresponding figure caption (where that kind of information belongs).

*P10 L30–P11 L2:* Same as previous point.

*Section 5:* The discussion covers mostly outlook-type points, whereas I think that potential weaknesses of the bias-adjusted data (including some of the criticism I've brought up) would need more discussion.

*Figures showing distribution parameters:* Many of these need units (e.g., the scale parameter in Fig. 1) which should be added to the colour bars.

*Figures showing distribution parameters:* Some colour bars should be adjusted to have green at zero to be more intuitive.

*Figures showing distribution parameters:* I can't resist mentioning that the used colour bars are not colour blindness friendly.

*Figure 18:* I would find it helpful if the symbol shape would still have the same meaning as in Fig. 17, but with colours coding the additional dimension of "bias improvement/degradation".

**Technical corrections**

*P2 L7–8 (and throughout the paper) – bias adjustment* versus *bias-adjustment:* Should be spelled consistently.

*All equations and lists:* These should be formatted with appropriate punctuation.

---

## Author Comment (AC1) · 16 Mar 2017

We thank both reviewers for the time taken in providing these thoughtful and extensive reviews.

Reviewer 1 made two principal points: i) a more thorough comparison of our bias-adjusted data with a reference dataset and ii) some issues with the way we bias adjusted dew point temperature. These will require some additional work and an alteration to the way we bias-adjust dewpoint temperature, taking the reviewer's suggestion as the starting point. Thus our revised paper will provide a comparison of our bias-adjusted approach and that of the WFDEI dataset, using air temperature and precipitation, comparing both to the E-OBS dataset over the 1979-2014 period (we had

already produced some diagrams of this type, but didn't include any in the paper). Here we have extended these to include the additional years of WFDEI and show some results at the end. We will also provide an alternate and more robust approach to bias-adjusting dewpoint temperature.

Reviewer 2 made points related to the potential independence of the observed station data assimilated into ERA-Interim with those used in the datasets we use to bias adjust ERA-Interim. We have addressed this by providing an additional section to the paper discussing the independence or lack thereof for some variables and some of the datasets used. Adding this section early in the paper will provide the reviewer with the necessary detail, but this requires more detail about the station and gridded datasets that we use and also the way station data are assimilated by ERA-Interim and other Reanalysis datasets. We couldn't find an appropriate source for this sort of detail/discussion. A Reanalysis could be considered as a data-infilling procedure, but it should be more-correctly considered as a physical/dynamic-infilling procedure, as opposed to a strict data-infilling procedure used, for example, by E-OBS, CRU TS and GPCC, which uses only the statistical properties of the specific variable. Indeed Re-analyses exploit the combination of observations and numerical models by incorporating additional information from multiple variables in such a way that a dynamical balance of all variables is maintained, while also considering their physical properties – this is achieved by imposing the standard mass and velocity fields, as well as energy, balances.

In the following the original reviews are in italics and our responses are in normal font. These different fonts can only be seen in the Supplement pdf, which also contains the additional difference plots discussed in this response.   Reviewer 1 General comments. Jones et al present a dataset of meteorological variables at half-degree resolution for Europe based on bias correction of ERA-Interim surface reanalysis data. They use observations to correct the distributions (e.g. mean and skewness) of the variables rather than just correcting the means. The new product is designed to be of

use for the energy sector.

Although well presented and clear, there are two overriding problems with the manuscript in its current form.

Firstly, their methodologies have limitations that are not explored in appropriate depth - particularly with regard to their revised distributions for near-surface air temperature, relative humidity and for daily precipitation totals.

Secondly, although the text repeatedly refers to an earlier dataset, also at half-degree resolution and based on bias correction of ERA-Interim reanalysis data (i.e. the WFDEI, Weedon et al. 2014 WRR) there is no comparison made between the new data and the existing dataset or other existing alternatives. Since they have corrected both the means and distributions of the ERA-Interim data I was surprised that there is no demonstration that compared to observations the new dataset represents an improvement over the existing datasets. This would both demonstrate that it is more useful for the energy sector and support their claim that their methodology is worth applying in other circumstances.

For the second point, we did intend making the comparisons with WFDEI, but we tried to limit the number of diagrams in the submitted version. To address this, we will include comparisons for air temperature and precipitation which are the two main climate variables used in many studies. They are also the two most well-measured variables across Europe.

As we had partly undertaken this comparison, we updated it to include the additional years of WFDEI mentioned by Reviewer 1. Some examples are given at the end of this response. As with our other plots, we illustrate differences for a pair of months and will provide the full twelve month set of plots via the same ftp site where all data and plots produced with this study are provided (for both air temperature and precipitation). We compare our bias adjustment and WFDEI separately against E-OBS based on averages over the 1979-2014 period. We chose E-OBS for both variables as this dataset

uses a much greater amount of basic station data than the CRU TS dataset for air temperature. This is also probably the case for precipitation when compared to GPCC, but we cannot verify this as GPCC does not reveal the locations and completeness of the station precipitation sources that are used. E-OBS for both variables is a gridded product on our standard grid, the same as the WFDEI dataset.

The addition of these plots will require some extra text to discuss the reasons for the differences. For most months for these two variables, and for this 36-year period, our adjusted ERA-Interim is closer to E-OBS. Differences are smoother for adjusted ERA-Interim and less spatially-variable (i.e. less spotty) than for WFDEI.

Specific comments.

a) Dew point temperature. The authors start the manuscript by stating that previous datasets applied bias corrections that were independent for each meteorological variable. This is misleading since the WATCH Forcing Data methodology (WFD), that was used to create the WFDEI (i.e. Weedon et al 2010 Tech Rep cited by Weedon et al 2011 JHM, Weedon et al 2014 WRR) specifically adopted the approach specified by Cosgrove et al 2003 (JGR). The Cosgrove et al approach was also used by another reanalysis-based dataset not mentioned: the Princeton Global Forcing (PGF) of Sheffield et al 2006 (J Clim). The Cosgrove et al approach ensures that changes to near-surface temperature are incorporated into changes in air pressure, specific humidity and then downwards longwave radiation by using sequential processing (maintaining the original covariances). We will refer to these additional datasets and correct any misleading statements. Surprisingly, despite raising this issue Jones et al adjusted air temperatures independently of dew point temperatures and consequently, as discussed on manuscript page 9, encountered implied relative humidities exceeding 100% (specifically avoided in the WFD, WFDEI and PGF). I strongly recommend that the authors design an approach that ensures that adjustments to air temperature and dew point temperature are applied in a consistent fashion - to avoid the possibility of their implied relative humidity exceeding 100%.

Dewpoint (DP) temperature was necessary to bias adjust Relative Humidity (RH), a variable required by our users. We discussed that the resulting RH values exceeding 100% arose due to our using E-OBS data for air temperature and HadISD for DP (i.e. we were not applying them in a consistent way, as the reviewer noted). In order to keep using the much more extensive (in terms of stations used) E-OBS dataset for air temperature, we will therefore apply a different approach. We have additionally extracted the HadISD dataset for air temperature. We already had the HadISD dataset for DP. The HadISD dataset is much less spatially extensive than E-OBS in terms of the number of near-complete stations used ($\sim$800 compared to $\sim$2000-3000 depending on the period for E-OBS).

Using both DP and air temperature (AT) data from HadISD (i.e. pairing off each individual 6-hour observation) we will calculate a dataset at each time step of Dewpoint Depression (DPD). DPD being AT minus DP, with DPD being greater than or equal to zero. To bias adjust ERA-Interim for DPD, we will additionally compute DPD from this Reanalysis. Then, we will bias adjust DPD ensuring AT $\geq$ DP. We will experiment with an appropriate distribution for DPD. As DPD is always positive or zero, possible distributions include the Weibull and Gamma Distribution.

b) Air temperature.

The Cosgrove et al (2003) methodology allows for elevation differences between observation sites and the mean elevation of reanalysis grid boxes. There are elevation differences between the heights of the EOBS temperature sites and the half degree elevations of the adjusted products. Hence an important step is to ensure that firstly the EOBS temperature data have been adjusted, using a lapse rate, to sea level, then interpolated to half degree and then recover the interpolated EOBS temperatures at the height of the ERA-Interim half degree grid boxes. As such the resulting interpolated EOBS data can then be used to correct the interpolated ERA Interim temperatures (similarly interpolated at sea level, Cosgrove et al 2003). This is such a critical processing step that I assume it was an oversight that it was not stated by the authors.

We will re-write the text for air temperature, to clarify that we are not using elevation fields as they are not needed. We admit that this aspect was not very clear in the original submitted paper. The reason that elevation fields are not needed is that both ERA-Interim and E-OBS have relatively smooth elevation fields (ERA-Interim at its native grid of $\sim0.7°$ by $0.7°$ and E-OBS at $0.5°$ by $0.5°$). These elevation fields are relatively similar to each other. A normal distribution is fit to each month (separately for the two datasets) for AT, based on the period 1981-2010. The bias-adjustment procedure is then applied to the full 1979-2014 period, using the distributional parameters from 1981-2010. Not using station elevations probably explains the smooth nature of the differences between adjusted ERA-Interim and E-OBS, as opposed to what is seen in our additional plots involving WFDEI.

We still have to designate a land mask for both datasets and these land masks differ somewhat around coastlines. We need to exclude air temperatures from ERA-Interim that result from grid boxes which are deemed to be ocean.

The average bias of their corrected air temperature product is shown for April in Figure 8. There are numerous areas within the map where the mean air temperatures do not match the observations to within a small part of a degree Celcius especially in mountainous areas and Turkey (where there is a bias of more than 2 oC). Since their methodology is designed to fix the means as part of the bias correction I find such differences to be surprising at the least and it definitely requires explicit explanation in the text. The text should also describe how and why the air temperature bias maps vary by month or season. Furthermore, the distributions of corrected air temperatures shown for the Scottish site differ substantially in terms of the most probable value by around one degree in winter (Dec and Jan). Somehow the modal temperature has been changed from more than the EOBS mode in the original ERA-Interim data to less than the EOBS mode.

Figure 8 is in essence a version of the additional plots we have produced (as it adjusted ERA-Interim minus E-OBS for the mean), but just for April for the 1981-2010 period.

[Figure]

The differences are within +/- 1 deg C for the majority of the domain, except for small regions in the Alps, northern Norway and the Balkans. Over a large portion of Turkey though, there are differences slightly larger than 2 deg C. This needs commenting upon, as it relates to poor availability (hardly any) of daily station temperature data from Turkey within the E-OBS dataset. We noted in our submitted paper that the quality and availability of daily data from North Africa, the near East and Turkey is poor. We will elaborate on this as although E-OBS is a WMO-Region-6-endorsed project, the supply of daily station data is uneven over the region, particularly over Turkey.

For the Kirkwall site, the modal distributions may differ in Figure 6, but this is not easily related to the overall mean. Our bias-adjustment procedure forces the means to agree, not the modal values. In the new plots of differences with E-OBS, all of the UK shows close relationships between the adjusted ERA-Interim and E-OBS. As an aside, all distributional plots (Figures 2, 3, 6, 7, 10, 11 and 14, 15) use the variables expressed in absolute terms.

I would have expected the modes to align everywhere after the adjustments. Again this requires specific comment. It is conceivable that the means do in fact match with say 0.1oC whereas the modes differ substantially for December and January, but this needs explanation/clarification for the reader. Are there similar issues elsewhere and if so why?

The new set of difference plots (Adjusted ERA-Interim and WFDEI versus E-OBS) show that the adjusted ERA-Interim for most of the domain (for March and April shown) is within the range +/- 1 with areas in northern Scandinavia, Turkey and parts of southern Europe occasionally showing values greater than 2 deg C for some months of the year (summer months more so than winter). In contrast, differences for WFDEI are more spatially variable and locally much larger, especially in mountainous regions and near some northern/western coasts. We will discuss these results more in the revised manuscript. The normal distribution we use for AT is not always a good fit in some winter months, when observed air temperatures tends to be negatively skewed. An

alternative distribution with an additional parameter is possible, but this would involve greater complexity that is probably not justified. These negatively skewed distributions occur elsewhere, but we provided all ∼4500 distributions for users to look at. We will comment more on this in the revised paper. A negative-skew in air temperatures is common in winter months in more continental regions. It is evident in monthly averages, but more prominent in daily data.

It is clearly difficult to relate the changes to the distribution to the overall averages for the 1979-2014 period. It is be much easier to look at the differences in the full set of difference plots which we provide.

c) Precipitation.

Aside from consideration of numbers of wet days, a gamma distribution has been used to correct the distribution of daily precipitation totals. Unfortunately, the illustrated results are not encouraging. For the Scottish example in Figure 14, in May to September the distribution is improved compared to the original data, but there appears to be room for further improvement. Nevertheless, illustration of the WFDEI and/or PGF results would probably put the adjusted Scottish values into a better perspective.

We pointed out in the paper that the gamma distribution although widely used is unlikely to be the perfect distribution for precipitation everywhere across Europe for all seasons. This is particularly the case for summer months in southern Europe where there can be too few wet days to fit any sort of distribution. It is necessary to have a common distribution everywhere to use any approach. A different distributional fit in the Scottish example might work better in summer, but the bias-adjusted ERA-Interim distribution moves the original ERA-Interim distribution closer to that of the station.

In the case from Slovenia (Figure 15) the results suggest significant methodological problems. In April to August it is clear that the adjustments have made the distributions worse than the original ERA Interim data. In these months not only is the modal value further from the observations than the raw data (green line further from black

obs than the orange line), but also at intermediate precipitation totals (2 to 8 mm/day) the corrected distribution is worse than the original data (in this region of the graphs higher).

In the original paper, we showed two examples for each variable. These were two examples from ∼800 for wind speed and DP and from ∼4520 for air temperature and precipitation. We did not go through all these examples and chose the most perfectly fitting ones. Instead, we showed the same two and altered the 3-month season for each of the four variables.

As with air temperature, there is an issue relating to the distributional fits at individual locations to the overall mean change over the 1979-2014 period. In the new plots of the differences between adjusted ERA-Interim and also WFDEI against E-OBS there are slightly greater differences for WFDEI. For Slovenia, it is again difficult to relate the distributional plots to the overall difference in means.

We will provide additional discussion of these new plots where differences for precipitation are more spatially varied (i.e. spotty) for WFDEI. Differences between adjusted ERA-Interim and WFDEI versus E-OBS in summer are similar in magnitude, but differences are much smaller for the adjusted ERA-Interim during the other seasons.

These problems should be discussed in the manuscript.

We will add more discussion, but there are also all the other plots that readers can access.

It is clear that the authors appreciate that different localities require different approaches (manuscript page 10 line 3). Why have the authors not designed code that, when the distribution is not improved by the adjustments the results are reverted to uncorrected values? Piani et al (2010 J Clim - NB not the Piani et al 2010 paper cited by Jones et al) adopted a variety of transfer functions for daily precipitation totals chosen according to their appropriateness by location (ranging from simple to reasonably

complex).

We pointed out that the Gamma Distribution is not perfect everywhere (see earlier response). We don't think it is right to change the precipitation distribution to another one just because the fit doesn't work that well on some occasions. If we did this, we wouldn't be able to apply the same type of adjustments across the whole domain for each month of the year.

We have corrected the Piani et al. (2010) reference.

The figure showing the spatial distribution of the adjusted precipitation results is provided for October (Figure 16). However, we are not told whether other months have worse performance (as is implied for the Slovenia example at least for April to August).

We provided ∼4500 precipitation distributional plots for readers to look at, as opposed to expecting users to calculate them from the original datasets. Figure 16 was for October, but we provided the other eleven months in the additional plots, which will be available via the CDS. As stated for the other variables, we didn't choose those that worked best. Any distributional choice is not going to work best everywhere. We will add some more discussion to emphasise this.

The manuscript definitely needs to appraise the success of the precipitation bias corrections more critically with reference to issues linked to both location and month or season.

d) Comparison to previous datasets. I am very surprised the authors did not explicitly demonstrate that their data provide advantages in terms of reliability compared to the existing, widely-used datasets that are currently available for the same time interval and the same spatial resolution (i.e. WFDEI and PGF). Both diagrams illustrating the point and a text discussion of the advantages of the new product merit a new section.

An additional set of plots comparing WFDEI and our bias-adjusted ERA-Interim with E-OBS have been calculated and added. At the end of this response, we provide some

examples.

Technical corrections.

p2 line 22 The WFDEI now extends to the end of 2014 (not 2013 as stated). We have used this update to 2014 in the above comparisons. Figure 14 There is no such thing as negative mm/day in precipitation. Regardless of the calculation of the gamma distribution extending to negative values, the plots should be truncated at 0.0 mm/day.

We have modified the distributional fits with respect to smoothness and they will be truncated to 0.0 mm/day. We have also described more details of how these distributional plots are produced.

Figures 17 and 18: a) Remove the grid lines (they are confusing and were not used on the other maps). b) Simplify the information by using a single symbol coloured according to the size of bias (this makes the maps more comparable to the previous way biases were shown). c) Illustrate a colour scale for easy determination of the size of the bias. d) Do not use green and red at the extremes of a distribution as it disadvantages colour blind readers. Simply use blue at the opposite extreme to red.

Figures 17 and 18 will be re-drawn using the same map projections and style as the rest of the maps. We will modify the colour bars used in many of the plots. We had been trying to use consistent scales for each variable and logical colour bars, which differ for each variable.

Recommendation: Major revisions addressing the concerns listed above. graham.weedon@metoffice.gov.uk, 8th Feb 2017. Interactive comment on Earth Syst. Sci. Data Discuss., doi:10.5194/essd-2016-67, 2017.   Reviewer 2 (Please note that I have not read RC1 before submitting my review to maintain independence.)

Thank you for this review and submitting it independently. Independence of reviewers is important and this is a potential problem with reviewers accessing each other's reviews. Not an issue here though.

The authors describe how they obtained a bias-adjusted dataset for energy-sector relevant parameters from the ERA-Interim reanalysis. I have a substantial number of remarks. Overall, I am not really convinced that the bias-adjustment as conducted here is a significant step forward regarding the quality of the data compared to the unadjusted data. However, I certainly think that the data set deserves being published and documented through this paper – if certain points are clarified and the quality of the results are discussed more extensively and critically. This pertains in particular to the required independence between the data used for bias adjustment and the data used to assess the pre- and post-adjustment quality. I therefore recommend to accept the paper subject to major revisions.

Thank you for the thoughtful comments. The paper will be revised in the light of both reviewers, and both comments have significantly improved the paper.

My main points of criticism include: 1. If I am not mistaken, only in one case – for surface solar irradiance – an independent data set is used to assess the effects of the bias-adjustment in a conclusive way. In all other cases it appears that the data used for the adjustment are then used to assess the post-adjustment "bias" – which is not exactly zero only because of certain interpolation/remapping steps. If I am wrong here, please explain your approach in a way that prevents other readers from repeating my misunderstanding.

ERA-Interim assimilates surface observational data, radiosondes and satellite data in a similar way as an operational weather forecast does. Altogether there are many millions of data being used in reanalysis. However, precipitation data are not assimilated, so these are independent. We have added a section on this in the background (see provisional text just before the difference plots). Reading Dee et al. (2011) and knowing what data is in the MARS archive at ECMWF, the amount of assimilated surface data for Europe is similar to what is in the HadISD dataset, as this data has the 6-hour time step that is required for assimilation. So the E-OBS dataset which uses daily Tx and Tn values are not assimilated. Some of these stations do also provide 6-hourly

time step data. There are 3-4 times more stations available to E-OBS than in HadISD.

2. It is also unclear to what extent the station and derived gridded data are independent of data that have entered the assimilation system used to generate the reanalysis in the first place, and what the impact of interdependence might be (e.g., for near-surface temperature).

See the answer to the previous point and the greater discussion we will add to the revised paper. Many other papers (e.g. the WFDEI papers we refer to whose lead author is Reviewer 1) are aware of these issues, but few papers provide an extensive summary with respect to independence. We will provide this.

3. It is stated that the figures showing the statistical data distributions (Figs. 2, 3, 6, 7, ...) exhibit the [corresponding] distribution fits. I think that this is not correct, but that the figures show the empirical data densities obtained by some smoothing, e.g., with a Gaussian filter. As a consequence, also those parameters that are positive by definition (wind speed and precipitation) exhibit non-zero densities at negative values (where the theoretical distribution fits should have zero density). This should be corrected in the text and also explained in the figure captions. Also, I suggest to add the two corresponding theoretical distributions to each of the plot panels (e.g., as dashed curves), so one can judge how well the empirical distributions are approximated by the theoretical distributions.

We will modify the Figures to exclude the apparent zero values for precipitation and wind speed (see also response to Reviewer 1). The distributions we are plotting are empirical distributional fits using the observed or ERA-Interim data. The distributional plots were smoothed approximations and this will be better explained.

There are only three distributions to show. There is the one for station observations, then the raw and adjusted ERA-Interim. All three are determined by empirical fits using well-used formulae. We did initially assess the goodness of fit of the Weibull (and other) distributions, but knowing that the vast majority were adequate was not very useful. We
provided all the individual station distribution fits for all variables.

Specific comments

Abstract: I suggest to mention in the abstract that univariate adjustment is applied. (To my understanding, multivariate approaches are mentioned in the paper several times, but were not applied to generate the present data set.)

The abstract will be modified to make this clearer. The abstract was probably too brief as well.

P1 L17–18 – The benefit of performing bias-adjustment is demonstrated by comparing initial and bias-adjusted ERA-Interim data against observations: See general comment 1.

The abstract will be expanded to include this point.

P1 L22–23 – These are reconstructions of past climates produced through the blending of observations with physical/numerical models which have been developed explicitly for climate monitoring and research: Speaking of the use of reanalyses for climate monitoring (which in my view implies the consideration of long-term trends), I recommend to add a comment of caution regarding possible spurious trends that can arise from changes in the observing system (e.g., Bengtsson et al. 2004 Can climate trends be calculated from reanalysis data?).

This is a good point, so a comment will be added, but with reference to a much more recent paper by Simmons et al (2017), where comparisons with more traditional approaches for large-scale temperature and precipitation averages have been compared.

P2 L5 – climate model: I think this should rather read weather (forecast) model, or maybe less specifically numerical model.

Climate Model is an overused term. We were using the terminology from Dee et al (2011) who produced ERA-Interim. For the UK, the operational weather forecasts use

exactly the same computer code for weather forecasts as for climate model simulation. ERA-Interim is a fixed version of the dynamical atmospheric model used for weather forecasting at ECMWF, as of ∼2010. We will clarify the terminology used.

P2 L5 – amount of observational data: Could be complemented to amount, type, and quality of observational data.

Maps are given of this on a global basis in Dee et al (2011). We refer to this. This is also related to the additional paragraph on the station data assimilated into ERA-Interim.

P2 L12–13 – the bias may be larger for [...] regions of sparse station coverage: If I understand correctly, this statement holds only if this refers to those stations from which the data enter the assimilation system. In this context a clear distinction should be made between assimilation-related stations and stations used for the bias adjustment. There is probably strong overlap, but I doubt that the two sets of stations are identical (not speaking of the various other data types that enter the assimilation system).

See earlier discussion and the new section discussing the overlap or not of the data entering ERA-Interim, E-OBS and also HadISD. It must also be remembered that the ERA-Interim assimilation system takes in much more additional data and gives different weights to each component.

P2 L31–32 – The benefit of performing bias-adjustment is demonstrated by comparing initial and bias-adjusted data against station observations and gridded observation products: See the general comment regarding independence of the data used to assess the bias-adjusted data.

See our response to this in the revised paragraph discussing independence.

Section 2: I would find it very helpful if a table that gives an overview over all the used data sets could be provided, stating if they are station data or gridded (and on what grids, originally and after interpolation), what their time resolution and period is, what they are used for in this paper (bias-adjustment versus independent assessment

versus ...), and so forth.

The various datasets could all be put into a Table. We will determine if this is helpful.

Section 2: I think the WRDC data should also be introduced in this section. The WRDC data was introduced in the irradiance section. We could also put this into Section 2, but the WRDC are somewhat different from the gridded products.

P3 L8–9 – This section provides details of ERA-Interim, and the various gridded and station observation datasets used to assess the quality of this Reanalysis: The various observation datasets are also the ones used for the bias-adjustment, right? That should be clarified.

This will be clarified. Maybe the Table will help? There was a list in Section 2.2?

P3 L9–10 – may be regridded and can be interpolated: It should be described only what has actually been done for this paper and associated data set, not what "may be done".

This was probably confusing here. It was something we considered but didn't follow through. This conditional sentence is probably easier removed.

P3 L14 – 3 h (forecast): Well, in this case every other time step is still an analysis, right?

This will be clarified. We were trying to provide a distinction between analysis and forecast variables within ERA-Interim. We have clarified this in the additional paragraph on data independence.

P3 L1–5: I don't find where CRU and GPCC are actually used in the remainder of the paper, except where it's stated that the authors "began by comparing ERA-Interim against the gridded observational products at the monthly timescale", followed by some statements that are, however, not supported by figures. It is stated that "they are of potential use" in certain circumstances, but it appears that they do not enter the
associated data set. If that's right, it should not be stated that these products are used (as done in P3 L24).

They were mentioned as they were other possible datasets that could have been used. This point will be clarified. GPCC, for example, potentially uses many more precipitation estimates than the other datasets, but GPCC only releases gridded products and not the raw input data.

Section 2.4 – HelioClim: I am wondering whether this satellite-derived data set might have been adjusted to the WRDC data – in which case the improvements shown in Fig. 8 might be not so conclusive? Please clarify.

A correcting table has been developed in 2015 between HelioClim-3 and ground data originating from several BSRN stations within the field-of-view of Meteosat for correcting 15 min estimates made by HelioClim-3. It has been established by merging all data, i.e. it is a global correction and not a local one. Inputs to the correcting table are the clearness index from HelioClim-3 and the solar zenith angle. There is no local input. In this respect, HelioClim is independent of the BSRN data set. The WRDC data set comprises 55 stations, of which six are BSRN stations. We will state the independency of the HelioClim-3 and the WRDC data set. Anyhow, the objective when using Helio-Clim was to obtain a gridded data set for adjusting ERA-Interim data because there are too few ground stations. We don't think that the fact that HelioClim is independent or not of WRDC data is a drawback in our approach.

Section 3: Given that, to my understanding, only univariate adjustment is applied to generate the present datasets, I find it distracting that much of this section is about methods for multivariate adjustment.

This will be modified in the text and only discussed at one point in the paper. If we had not mentioned multivariate adjustments, we would have had comments about this.

P5 L8 – Bias adjustment and bias correction are widely-used terms for the assessment

of climate model output: One can assess climate model output without having bias adjustment in mind.

Any use of climate model output should always bear in mind that this output should be assessed for bias. We have probably overused the word 'assess' and will used a more specific verb.

P5 L11–12 – the assumption is made there that Reanalyses are correct: I don't think that anyone assumes absolute correctness of reanalysis data – which can equally not be said of station data.

The point we were making here is that Reanalysis data have been used to bias adjust climate model data in regions of sparse observational coverage. We will add a reference where this has been undertaken in Africa.

P6 L26 – for each month: I guess this is meant in a climatological sense, right? So I recommend to modify this to something like for each month of the year. Text will be modified.

P7 L4–6 – The similarity of the two distributions in terms of their scale and shape parameters indicates that bias adjustment could be achieved by replacing the ERAInterim scale and shape parameters with those inferred from the HadISD stations: Im not convinced by this argument. If a transformation based on a certain distribution type is "valid" should in my view rather depend on the degree to which the two samples are consistent with the theoretical distribution used. The fact that the parameters are similar rather indicates that the bias is not very large, after all.

This piece of text will be clarified. We are using empirical fits to the various distributions we use.

P7 L12–13 – Figures 2 and 3 exhibit the Weibull distribution fits of the HadISD observations: I think that this is not correct, see my corresponding general comment. Text will be clarified.

P7 L26–27 – the number of stations in some parts of Europe is less dense, so involving greater extrapolation from stations more distant from the grid boxes: I fail to understand why this should result in larger discrepancies in less dense areas. After all, locations with no stations close to them are not evaluated – because there are no stations at that location to do that! Shouldn't it be quite the opposite, namely, that isolated stations should show only small discrepancies after the procedure because the nearest grid box is influenced almost exclusively by that station, implying very weak interpolation effects? Please clarify. (This point is closely related to my general comment regarding missing independence of data used for adjustment and (the same) data used to check the post-adjustment "bias".)

This depends on how the station data in assimilated and used within ERA-Interim. This aspect will be partly covered by the additional paragraph on independence of different datasets. How well a Reanalysis simulates an isolated station depends on how much influence that station has. A Reanalysis is not a data smoother/interpolator. The E-OBS dataset is a gridded dataset where a sparsely located station will have a greater influence than one in a dense region.

P8 L3 – for the nearest E-OBS grid box: Is E-OBS not on the same grid as the interpolated ERA-Interim data? Please clarify.

E-OBS and our interpolated ERA-Interim are on the same 0.5 by 0.5 degree lat/long grid.

P8 L4 – Data are then normalized as in Equations 4 and 5: It appears that the normalization is described only be Eq. 4, whereas Eq. 5 describes the back-transformation.

Text will be clarified.

P9 L8–11 – Any type of bias adjustment procedure will additionally be influenced by the quality of the station observations, [e.g.,] by the large differences in potential height between some observing locations and the average height field used by ERA-Interim:

Indeed, and I think this point – stations at a particular point not necessarily being representative of a "grid cell" – should be mentioned more prominently and generally in the paper (it doesn't hold just for temperature and humidity, but for all considered parameters).

This will be addressed in the revised paragraph about independence and representativeness of stations versus grid-box averages.

P9 Equation 6: I think that the right-hand-side gives the probability density function rather than the cumulative distribution function (which the left-hand-side implies). In addition, it appears that is the scale parameter and the shape parameter, rather than the other way around as stated in the subsequent sentence. This will be checked and revised.

P9 L24–25 – all precipitation amounts below the threshold are set to zero, further improving the agreement between E-OBS and ERA-Interim in the number of dry days per month (i.e. days with rainfall less than the 1.0 mm threshold): This seems illogical: the modified values were already below the threshold, so they were already "dry days" according to the definition.

This depends on what users do with the data. Another user may have a different definition of a wet day. We just chose a commonly used one. We will clarify the text here. As the precipitation fits only consider the wet days, we need to set the low precipitation amounts to zero precipitation.

P10 L17–18 – The method 'quantile mapping' applied to KT was preferred (and is used here) as it usually brings improvement with no degradation of the bias, while the other methods often degrade the bias in a noticeable way: What does this imply for the other parameters where no independent validation was conducted, and where only on adjustment method was applied? It appears to me that this point to some extent compromises the validity of the bias-adjustment approach used in this work.

The independence of the other variables has been discussed earlier. Precipitation data are completely independent. Additionally, the new set of maps shows that as a monthly total, the bias adjustments are relatively small.

P10 L21–23: Half of this paragraph is just a repetition of the corresponding figure caption (where that kind of information belongs).

We have been careful to avoid such repetitions. We will check this.

P10 L30–P11 L2: Same as previous point.

See previous point.

Section 5: The discussion covers mostly outlook-type points, whereas I think that potential weaknesses of the bias-adjusted data (including some of the criticism I've brought up) would need more discussion.

We will add some more discussion here.

Figures showing distribution parameters: Many of these need units (e.g., the scale parameter in Fig. 1) which should be added to the colour bars.

Some of the distribution plots are dimensionless, so we will modify the Figure Captions accordingly.

Figures showing distribution parameters: Some colour bars should be adjusted to have green at zero to be more intuitive.

We will reassess the distribution plots, especially with respect to their scaling parameters. Figures showing distribution parameters: I can't resist mentioning that the used colour bars are not colour blindness friendly.

We will re-assess all the plots with colour bars, in the light of colour-blindness issues.

Figure 18: I would find it helpful if the symbol shape would still have the same meaning as in Fig. 17, but with colours coding the additional dimension of "bias improvement/

degradation".

Figures 17 and 18 will be redrawn in a similar way to the other spatial maps. We will also improve the explanation of the symbols used.

Technical corrections

P2 L7–8 (and throughout the paper) – bias adjustment versus bias-adjustment: Should be spelled consistently.

When used as a noun, the term is 'bias adjustment', but when as an adjective the hyphen is added, so 'bias-adjustment techniques'.

All equations and lists: These should be formatted with appropriate punctuation.

We will modify the equations with appropriate punctuation.

Independence of the station/gridded observation series versus ERA-Interim To likely go between sections 2 and 3 of the paper.

In this study, we propose bias-adjusting ERA-Interim for the 5 ECVs (wind speed, air temperature, dewpoint temperature, precipitation and irradiance). As stated in Section 2.1, ERA-Interim assimilates many different climate datasets in the development of the latest Reanalysis product (surface station data is just one of several; satellite and radiosonde data are also assimilated). In this section, we discuss how independent the station observations and gridded products are that are used in this bias adjustment compared to the surface station data assimilated into ERA-Interim. Our precipitation and irradiance data are totally independent as these data are not assimilated. These variables are forecast outputs from ERA-Interim (see Dee et al., 2011). Of the other three variables air and dewpoint temperatures are assimilated. For wind, the u and v components of the wind are assimilated. It is important to understand what is assimilated and what importance may be given to these variables. The output for these three variables we use is their value in the analysis (referred to as an analysis variable), produced every 6 hours. ERA-Interim doesn't provide details of all the specific station data (and additional satellite and radiosonde data) that are assimilated. Dee et al (2011) give details of what datasets are available for assimilation. ERA-Interim provides a dynamically consistent estimate of the climate state at each 6-hour time step, but it doesn't specifically give any details of which potential information was used to produce the analysis variables. Through dynamical consistency, information from satellites, radiosondes and other surface variables (e.g. pressure) are also used. In the subsequent version of ERA-Interim (ERA-5) due for completion by the end of 2018, an Observational Feedback Archive (OFA) is planned, but this currently doesn't exist. What ERA-Interim assimilates is values 4 times per day at the fixed 6-hour time steps (00, 06, 12 and 18). Essentially, the quantity of these data are similar to those available in the HadISD database, which we know are about 1500 series, but only about 800 are complete enough over the 1979-2014 period (Dunn et al., 2014). Thus for air temperature, the many thousands of additional daily Tx and Tn observations are not assimilated. So our E-OBS dataset for air temperature contains a much greater volume of additional temperature series than assimilated within ERA-Interim. The wind speed and dewpoint temperature from HadISD should have been available for assimilation, but the importance given to these observations is not as great as the importance given to the station pressure observations. The production of a Reanalysis has occasionally been referred to as dynamic infilling, which is quite different from the spatial infilling techniques that are used to produce the E-OBS, CRU TS and GPCC datasets. Spatial infilling techniques use a variety of statistical procedures (e.g. inverse distance weighting and kriging) and are generally applied for each variable independent of other variables. In data sparse regions, statistically infilling techniques will likely spread information from the few available stations across the unobserved areas. The effects of this are generally evident as reduced variance in the generated fields. In contrast, a Reanalysis will make use of additional information (e.g. the large-scale circulation) and

potentially not placing great emphasis on specific station data in a sparsely-observed region. In addition, balances of mass, wind and energy fields means that consistency between different variables is ensured, though this is particularly the case for forecast variables at a few-to-several hours lead time. At analysis time, such balance might be not guaranteed, but this depends on the specific data assimilation scheme used and whether the scheme enforces physical/dynamical balances.

Additional References Simmons, A. J., Berrisford, P., Dee, D. P., Hersbach, H., Hirahara, S. and Thépaut, J.-N. (2017), A reassessment of temperature variations and trends from global reanalyses and monthly surface climatological datasets. Q.J.R. Meteorol. Soc., 143: 101–119. doi:10.1002/qj.2949

Example Difference Maps

Air Temperature (for March and April)

First pair, adjusted ERA-Interim minus EOBS, then WFDEI minus EOBS.

Precipitation Totals (for September and October)

First pair, adjusted ERA-Interim minus EOBS, then WFDEI minus EOBS.

These chosen as we looked at April for Air Temperature earlier and October for Precipitation

Adjusted ERA-Interim is always closer to EOBS for air temperature than WFDEI for the 1979-2014 period. This is also the case for precipitation, but WFDEI is similar or slightly better during JJA, but poorer (more colour on the maps) in the other months.

As stated at the beginning, these different fonts can only be seen in the Supplement pdf, which also contains the additional difference plots discussed in this response.

Please also note the supplement to this comment:
http://www.earth-syst-sci-data-discuss.net/essd-2016-67/essd-2016-67-AC1-

supplement.pdf